# Learning Viewpoint-Agnostic Visual Representations by Recovering Tokens in 3D Space

**Jinghuan Shang**[1]    **Srijan Das**[1,2]    **Michael S. Ryoo**[1]

Department of Computer Science

[1]Stony Brook University, [2]University of North Carolina at Charlotte

[1]{jishang, mryoo}@cs.stonybrook.edu, [2]sdas24@uncc.edu

## Abstract

Humans are remarkably flexible in understanding viewpoint changes due to visual cortex supporting the perception of 3D structure. In contrast, most of the computer vision models that learn visual representation from a pool of 2D images often fail to generalize over novel camera viewpoints. Recently, the vision architectures have shifted towards convolution-free architectures, visual Transformers, which operate on tokens derived from image patches. However, these Transformers do not perform explicit operations to learn viewpoint-agnostic representation for visual understanding. To this end, we propose a 3D Token Representation Layer (3DTRL) that estimates the 3D positional information of the visual tokens and leverages it for learning viewpoint-agnostic representations. The key elements of 3DTRL include a pseudo-depth estimator and a learned camera matrix to impose geometric transformations on the tokens, trained in an unsupervised fashion. These enable 3DTRL to recover the 3D positional information of the tokens from 2D patches. In practice, 3DTRL is easily plugged-in into a Transformer. Our experiments demonstrate the effectiveness of 3DTRL in many vision tasks including image classification, multi-view video alignment, and action recognition. The models with 3DTRL outperform their backbone Transformers in all the tasks with minimal added computation. Our code is available at https://github.com/elicassion/3DTRL.

## 1 Introduction

Over the past few years, computer vision models have developed rapidly from CNNs [6, 26, 58] to now Transformers [16, 24, 59]. With these models, we can now accurately classify objects in an image, align image frames among video pairs, classify actions in videos, and more. Despite their successes, many of the models neglect that the world is in 3D and do not extend beyond the XY image plane [20]. While humans can readily estimate the 3D structure of a scene from 2D pixels of an image, most of the existing vision models with 2D images do not take the 3D structure of the world into consideration. This is one of the reasons why humans are able to recognize objects in images and actions in videos regardless of their viewpoint, but the vision models often fail to generalize over novel viewpoints [11, 20, 42].

Consequently, in this paper, we develop an approach to learn viewpoint-agnostic representations for a robust understanding of the visual data. Naive solutions to obtain viewpoint-agnostic representation would be either supervising the model with densely annotated 3D data, or learning representation from a large scale 2D datasets with samples encompassing different viewpoints. Given the fact that such high quality data are expensive to acquire and hard to scale, an approach with a higher sample efficiency without 3D supervision is desired.

To this end, we propose a 3D Token Representation Layer (3DTRL), incorporating 3D camera transformations into the recent successful visual Transformers [7, 16, 36, 59]. 3DTRL first recovers camera-centered 3D coordinates of each token by depth estimation. Then 3DTRL estimates a camera

36th Conference on Neural Information Processing Systems (NeurIPS 2022).

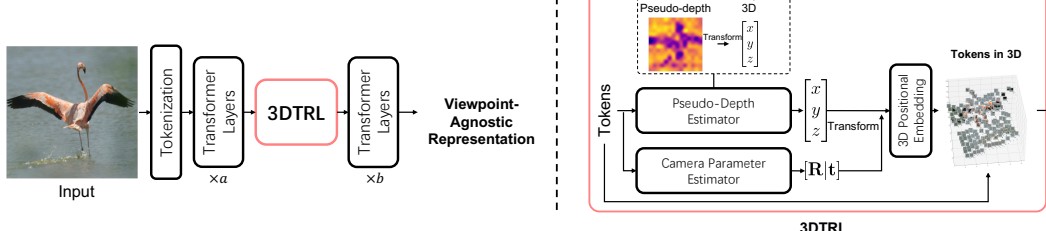

Figure 1: Overview of proposed 3DTRL. **Left**: 3DTRL is a module inserted in between Transformer layers. **Right**: 3DTRL has three parts, Pseudo-Depth Estimator, Camera Parameter Estimator, and 3D Positional Embedding layer. Within Pseudo-Depth Estimator, we first estimate depth of each token and then calculate 3D coordinates from 2D locations and depth.

matrix to transform these camera-centered coordinates to a 3D world space. In this world space, 3D locations of the tokens are absolute and view-invariant, which contain important information for learning viewpoint-agnostic representations. Therefore, 3DTRL incorporates such 3D positional information in the Transformer backbone in the form of 3D positional embeddings, and generates output tokens with the 3D information. Unlike visual Transformers only relying on 2D positional embedding, models with 3DTRL are more compliant with learning viewpoint-agnostic representations.

We conduct extensive experiments on various vision tasks to confirm the effectiveness of 3DTRL. Our 3DTRL outperforms the Transformer backbones on 3 image classification, 5 multi-view video alignment, and 2 multi-view action recognition datasets in their respective tasks. Moreover, 3DTRL is a light-weighted, plug-and-play module that achieves the above improvements with minimal (2% computation and 4% parameters) overhead.

In summary, we present a learnable, differentiable layer 3DTRL that efficiently and effectively learns viewpoint-agnostic representations.

## 2  Background: Pinhole Camera Model

3DTRL is based on the standard pinhole camera model widely used in computer vision. Thus, we first briefly review the pinhole camera model. In homogeneous coordinate system, given a point with world coordinate $p^{\text{world}}$, a camera projects a pixel at $p$ on an image by:

$$p = \mathbf{K} \left[\, \mathbf{R} | \mathbf{t} \,\right] p^{\text{world}}, \tag{1}$$

where $\mathbf{K}$ is the intrinsic matrix and $\left[\, \mathbf{R} | \mathbf{t} \,\right]$ is the extrinsic matrix. $\mathbf{K}$ is further represented by

$$\mathbf{K} = \begin{bmatrix} c & 0 & u_0 \\ 0 & c & v_0 \\ 0 & 0 & 1 \end{bmatrix}, \tag{2}$$

where $c$ is the focal length and $(u_0, v_0)$ are offset. In this work, we explore visual understanding in multi-view setting, thus aiming at learning viewpoint-agnostic representations. In this setting, a scene may be captured with different cameras positioned at non-identical locations and viewing angles. Here, the world coordinate $p^{\text{world}}$ is the same across all the cameras while pixel projection $p$ is different across cameras. We focus on how to estimate the world coordinates $p^{\text{world}}$ from their corresponding image pixels at $p$. Estimating $p^{\text{world}}$ from $p$ involves two transformations that correspond to the *inverse* of $\mathbf{K}$ and $\left[\, \mathbf{R} | \mathbf{t} \,\right]$ which might not be known beforehand. Thus, we learn to *estimate* them from image patches instead, which is a key procedure in 3DTRL.

## 3  3D Token Representation Layer (3DTRL)

In this section, we detail how 3DTRL estimates 3D positional information in a Transformer and is integrated. We will introduce 3DTRL for image analysis, then we adapt it to video models.

### 3.1  Overview

3DTRL is a simple yet effective plug-in module that can be inserted in between the layers of visual Transformers (Figure 1 left). Given a set of input tokens $S$, 3DTRL returns a set of tokens with 3D information. The number of tokens and their dimensionality are kept unchanged. Within the

module (Figure 1 right), 3DTRL first performs 3D estimation using two estimators: (1) pseudo-depth estimator and (2) camera parameter estimator. Then the tokens are associated with their recovered world 3D coordinates, which are transformed from estimated depth and camera matrix. Finally, 3DTRL generate 3D positional embeddings from these world coordinates and combine them with input $S$ to generate the output of 3DTRL.

In the aspect that we insert 3DTRL in between the Transformer model, 3DTRL implicitly leverages the Transformer layers *before* it to be a part of the 3D estimators, and layers *after* that to be the actual 3D feature encoder (Figure 1 left). This avoids adding a large number of parameters while resulting in reasonable estimations. We empirically find that placing 3DTRL at a shallow-medium layer of the network yields better results (see Section 4.6), which is a trade-off between model capacity for estimation and 3D feature encoding.

## 3.2 3D Estimation of the Input Tokens

3DTRL first estimates both the camera-centered 3D coordinates of each token using depth estimation and a camera matrix shared by all the tokens of an image. Then, the camera-centered 3D coordinates are transformed to the "world" coordinates by the camera matrix.

**Pseudo-depth Estimation.** Given input tokens $S = \{s_1, \ldots, s_N\} \in \mathbb{R}^{N \times m}$, 3DTRL first performs pseudo-depth estimation of each token $s_n$. The pseudo-depth estimator is a function $f : \mathbb{R}^m \to \mathbb{R}$ that outputs the depth $d_n = f(s_n)$ of each token individually. We implement $f$ using a 2-layer MLP. We call this pseudo-depth estimation since it is similar to depth estimation from monocular images but operates at a very coarse scale, given that each token corresponds to an image patch rather than a single pixel in Transformer.

After pseudo-depth estimation, 3DTRL transforms the pseudo-depth map to camera-centered 3D coordinates. Recall that in ViT [16], an image $X$ is decomposed into $N$ patches $\{X_1, \ldots, X_N\} \in \mathbb{R}^{N \times P \times P \times 3}$, where $P \times P$ is the size of each image patch and the tokens $S$ are obtained from a linear projection of these image patches. Thus, each token is initially associated with a 2D location on the image plane, denoted as $(u, v)$. By depth estimation, 3DTRL associates each token with one more value $d$. Based on the pinhole camera model explained in Section 2, 3DTRL transforms $u, v, d$ to a camera-centered 3D coordinate $(x, y, z)$ by:

$$p_n^{\text{cam}} = \begin{bmatrix} x_n \\ y_n \\ z_n \end{bmatrix} = \begin{bmatrix} u_n z_n / c \\ v_n z_n / c \\ z_n \end{bmatrix}, \text{ where } z_n = d_n. \tag{3}$$

Since we purely perform the aforementioned estimation from monocular images and the camera intrinsic matrix is unknown, we simply set $c$ to a constant hyperparameter. To define coordinate system of 2D image plane $(u, v)$, we set the center of the original image is the origin $(0, 0)$ for convenience, so that the image plane and camera coordinate system shares the same origin. We use the center of the image patch to represent its associate $(u, v)$ coordinates.

We believe that this depth-based 3D coordinate estimation best leverages the known 2D geometry, which is beneficial for later representation learning. We later confirm this in our ablation study (in Section 4.6), where we compare it against a variant of directly estimating $(x, y, z)$ instead of depth.

**Camera Parameter Estimation.** The camera parameters are required to transform the estimated camera-centered 3D coordinates $p^{\text{cam}}$ to the world coordinate system. These camera parameters are estimated jointly from all input tokens $S$. This involves estimation of two matrices, a $3 \times 3$ rotation matrix $\mathbf{R}$ and a $3 \times 1$ and translation matrix $\mathbf{t}$ through an estimator $g$. We implement $g$ using a MLP. Specifically, we use a shared MLP stem to aggregate all the tokens into an intermediate representation. Then, we use two separated fully connected heads to estimate the parameters in $\mathbf{R}$ and $\mathbf{t}$ respectively. We note the camera parameter estimator as a whole: $[\mathbf{R}|\mathbf{t}] = g(S)$. To ensure $\mathbf{R}$ is mathematically valid, we first estimate the three values corresponding to yaw, pitch and roll angles of the camera pose, and then convert them into a $3 \times 3$ rotation matrix. Research has shown that the discontinuity occurs at the boundary cases in rotation representation [63, 69], however, such corner cases are rare.

In case of generic visual understanding tasks like object classifications, we expect the camera parameter estimation to perform an "object-centric canonicalization" of images with respect to the "common" poses of the class object. This is qualitatively shown by Figure 10 in the Appendix.

**Transform to World Coordinates.** Now, with the camera parameters, 3DTRL transforms estimated camera-centered coordinates $p^{\text{cam}}$ into the world space, a 3D space where 3D coordinates of the

tokens are absolute and viewpoint-invariant. Following the pinhole camera model, we recover $p^{\text{world}}$:

$$p_n^{\text{world}} = [\ \mathbf{R}^T | \mathbf{R}^T \mathbf{t}\ ]\ p_n^{\text{cam}}. \tag{4}$$

## 3.3 Incorporating 3D Positional Information in Transformers

The last step of 3DTRL is to leverage the estimated 3D positional information in Transformer backbone. For this, we choose to adopt a typical technique of incorporating positional embedding that is already used in Transformers [16, 60]. In contrast to 2D positional embedding in ViTs [16], 3DTRL learns a 3D embedding function $h : \mathbb{R}^3 \rightarrow \mathbb{R}^m$ to transform estimated world coordinates $p^{\text{world}}$ to positional embeddings $p^{\text{3D}}$. This 3D embedding function $h$ is implemented using a two-layer MLP. Then, the obtained 3D positional embedding is incorporated in the Transformer backbone by combining it with the token representations. The outcome is the final token representations $\{s^{\text{3D}}\}$:

$$s_n^{\text{3D}} = s_n + p_n^{\text{3D}}, \text{ where } p^{\text{3D}} = h(p^{\text{world}}). \tag{5}$$

After 3D embedding, the resultant token representations are associated with a 3D space, thus enabling the remaining Transformer layers to encode viewpoint-agnostic token representations. We ablate other ways of incorporating the 3D positional information of the tokens in Section 4.6.

## 3.4 3DTRL in Video Models

Notably, 3DTRL can be also easily generalized to video models. For video models, the input to 3DTRL is a set of spatial-temporal tokens $\{S_1, \ldots, S_T\}$ corresponding to a video clip containing $T$ frames, where $S_t = \{s_{t1}, \ldots, s_{tN}\}$ are $N$ tokens from $t$-th frame. We simply extend our module to operate on an additional time dimension, where depth estimation and 3D positional embedding are done for each spatial-temporal tokens $s_{tn}$ individually: $d_{tn} = f(s_{tn})$, $p_{tn}^{\text{3D}} = h(p_{tn}^{\text{world}})$. Camera parameters are estimated per input frame ($S_t$) in a dissociated manner, namely $[\ \mathbf{R}|\mathbf{t}\ ]_t = g(S_t)$, resulting in a total of $T$ camera matrices per video. We investigate another strategy of camera estimation in the supplementary, where only one camera matrix is learned for all frames. However, our studies have substantiated the effectiveness of learning dissociated camera matrices per frame.

# 4 Experiments

We conduct extensive experiments to demonstrate the efficacy of viewpoint-agnostic representations learned by 3DTRL in multiple vision tasks: (i) image classification, (ii) multi-view video alignment, and (iii) video action classification. We also qualitatively evaluate the pseudo-depth and camera estimation to confirm 3DTRL works.

## 4.1 Image Classification

In order to validate the power of 3DTRL, we first evaluate ViT [16] with 3DTRL for image classification task using CIFAR-10 [34], CIFAR-100 [34] and ImageNet-1K [15] datasets.

**Training**  We use the training recipe of DeiT [59] for training our baseline Vision Transformer model on CIFAR and ImageNet datasets from *scratch*. We performed ablations to find an optimal location in ViTs where 3DTRL should be plugged-in (Section 4.6). Thus, in all our experiments we place 3DTRL after 4 Transformer layers, unless otherwise stated. The configuration of our DeiT-T, DeiT-S, and DeiT-B is identical to that mentioned in [59]. All our transformer models are trained for 50 and 300 epochs for CIFAR and ImageNet respectively. Further training details and hyper-parameters can be found in the supplementary material.

**Results**  From Table 1, 3DTRL with all DeiT variants shows consistent performance improvement over the baseline on the CIFAR datasets, with only ∼2% computation overhead and ∼4% more parameters. Despite fewer training samples, 3DTRL significantly outperforms the baseline in CIFAR, showing the strong generalizability of 3DTRL when limited training data are available. We argue that multi-view data is not available in abundance, especially in domains with limited data (like in medical domain), thus 3DTRL with its ability to learn viewpoint-agnostic representations will be crucial in such domains. We also find the performance improvement on ImageNet is less than that on CIFAR. This is because ImageNet has limited viewpoints in both training and validation splits, thus reducing the significance of performing geometric aware transformations for learning view agnostic representations. In contrast, CIFAR samples present more diverse camera viewpoints, so it is a more suitable dataset for testing the quality of learned viewpoint-agnostic representations.

Table 1: Top 1 classification accuracy (%) on CIFAR-10 and 100, ImageNet-1K (IN-1K), viewpoint-perturbed IN-1K (IN-1K-*p*), and ObjectNet. We also report the number of parameters (#params) and computation (in MACs). Note that the MACs are reported w.r.t. IN-1K samples.

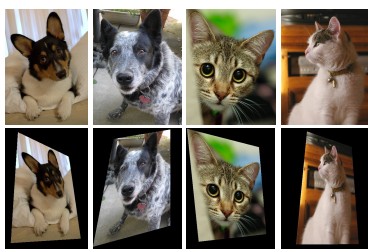

| Method | #params | MACs | CIFAR-10 | CIFAR-100 | IN-1K | IN-1K-*p* | ObjectNet |
|--------|---------|------|----------|-----------|-------|-----------|-----------|
| DeiT-T | 5.72M | 1.08G | 74.1 | 51.3 | 73.4 | 61.3 | 21.3 |
| +3DTRL | 5.95M | 1.10G | **78.8** (+4.7) | **53.7** (+2.4) | **73.6** (+0.2) | **64.6** (+3.3) | **22.4** (+1.1) |
| DeiT-S | 22.1M | 4.24G | 77.2 | 54.6 | 79.4 | 71.1 | 25.8 |
| +3DTRL | 23.0M | 4.33G | **80.7** (+3.5) | **61.5** (+6.9) | **79.7** (+0.3) | **72.7** (+1.6) | **27.1** (+1.3) |
| DeiT-B | 86.6M | 16.7G | 76.6 | 51.9 | 81.0 | 70.6 | 27.0 |
| +3DTRL | 90.1M | 17.2G | **82.8** (+6.2) | **61.8** (+9.9) | **81.2** (+0.2) | **74.7** (+4.1) | **27.3** (+0.3) |

Figure 2: **Top**: original IN-1K samples. **Bottom**: viewpoint-perturbed IN-1K samples.

**Robustness on Data with Viewpoint Changes**   In order to emphasize the need of learning viewpoint-agnostic representations, we further test the models trained on IN-1K on two test datasets: ObjectNet [2] and ImageNet-1K-perturbed (IN-1K-*p*). ObjectNet [2] is a *test* set designed to introduce more rotation, viewpoint, and background variances in samples compared to ImageNet. Overall, ObjectNet is a very challenging dataset considering large variance in real-world distributions. IN-1K-*p* is a *viewpoint*-perturbed IN-1K validation set by applying random perspective transformations to images, constructed by our own. Example images are shown in Figure 2. We note that perspective transformation on these static images is not equivalent to real viewpoint changes. Nonetheless, it is a meaningful for a proof-of-concept experiment. Results in Table 1 show that models with 3DTRL consistently outperform their corresponding Transformer baselines in these two test sets, suggesting 3DTRL is trained to generalize across viewpoint changes.

## 4.2   Multi-view Video Alignment

Video alignment [18, 23, 41] is a task to learn a frame-to-frame mapping between video pairs with close semantic embeddings. In particular, we consider a *multi-view* setting that aligns videos captured from the same event but different viewpoints, which could further facilitate robot imitation learning from third-person views [4, 49, 51, 56]. Here, video pairs from the same event are temporally synchronized.

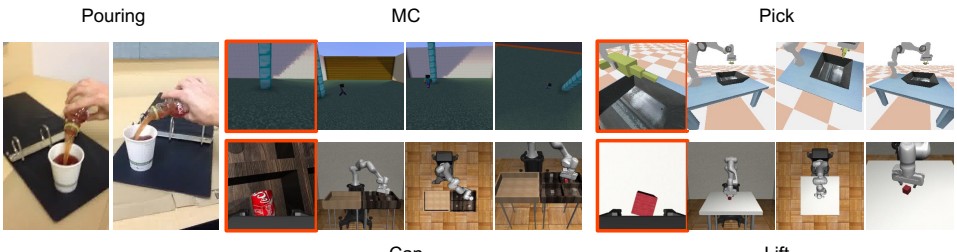

Figure 3: Examples video alignment datasets. Each dataset has synchronized videos of at least 2 viewpoints. All datasets except Pouring have one ego-centric view, highlighted in red boxes. More details are available in supplementary material.

**Datasets**   We use 5 multi-view datasets from a wide range of environments: **Minecraft** (MC) – video game, **Pick**, **Can**, and **Lift** from robot simulators (PyBullet [9] and Robomimic [39]), and **Pouring** from real-world human actions [49]. Example video frames are provided in Figure 3. Each dataset contains synchronized videos from multiple cameras (viewpoints). There is one ego-centric camera per dataset except Pouring, which are continuously moving with the subject. These ego-centric videos make the alignment more challenging. Detail dataset statistics are available in supplementary.

**Training**   We follow common video alignment methods [49] to train an encoder that outputs frame-wise embeddings. We still use DeiT [59] as a baseline model (DeiT+TCN) and apply 3DTRL to it (+3DTRL), similar to image classification. During training, we use the time-contrastive loss [49] to encourage temporally closed embeddings to be similar while temporally far-away embeddings to be apart. Then, we obtain alignments via nearest-neighbor such that an embedding $u_i$ from video 1 is being paired to its nearest neighbor $v_j$ in video 2 in the embedding space. And similarly $u_j$ from video 2 is paired with its nearest neighbor $v_k$ in video 1. We use ImageNet-1K pre-trained weights for experiments on Pouring, but we train from scratch for other datasets considering that simulation environments are out of real-world distribution.

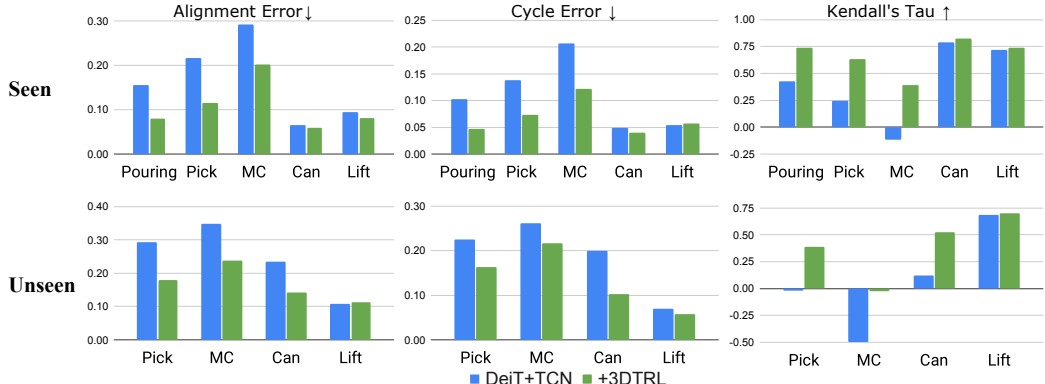

Figure 4: Results on video alignment in **Seen** and **Unseen** protocols. ↑ indicates a higher metric is better and ↓ is otherwise. Blue bars are for DeiT+TCN without 3DTRL and green bars are with 3DTRL. 3DTRL outperforms the baseline consistently in both settings. We note **Unseen** is not applicable to Pouring dataset because only two cameras are available in this dataset.

**Evaluation**    We evaluate the alignment by three metrics: Alignment Error [49], Cycle Error, and Kendall's Tau [33]. Let the alignment pairs between two videos be $(u_i, v_j)$ and $(v_j, u_k)$. In brief, Alignment Error measures the temporal mismatching $|i - j|$ of $(u_i, v_j)$. Cycle Error is based on cycle-consistency [18, 62], where two pairs $(u_i, v_j)$ and $(v_j, u_k)$ are called *consistent* when $i = k$. Thus, Cycle Error measures the inconsistency based on distance metric $|i - k|$. Kendall's Tau ($\tau$) measures ordering in pairs. Given a pair of embeddings from video 1 $(u_i, u_j)$ and their corresponding nearest neighbors from video 2 $(v_p, v_q)$, the indices tuple $(i, j, p, q)$ is *concordant* when $i < j$ and $p < q$ or $i > j$ and $p > q$. Otherwise the tuple is *discordant*. Kendall's Tau computes the ratio of concordant pairs and discordant pairs over all pairs of frames. Let $N$ be the number of frames in a video, then the formal notations of the three metrics are:

$$\text{Alignment Error} = \mathbb{E}_i \frac{|i-j|}{N}; \ \text{Cycle Error} = \mathbb{E}_i \frac{|i-k|}{N}; \ \tau = \frac{\#\text{ concordant pairs} - \#\text{ discordant pairs}}{N(N-1)/2}. \tag{6}$$

We establish two evaluation protocols: (a) **Seen** and (b) **Unseen**. In **Seen**, we train and test models on videos from all cameras. However, in **Unseen**, we hold out several cameras for test, which is a representative scenario for validating the effectiveness of 3DTRL. Detail of experimental settings are provided in supplementary.

**Results**    Figure 4 illustrates the evaluation results of 2 viewpoint settings over 5 datasets, compared to the DeiT baseline. 3DTRL outperforms the baseline consistently across all datasets. In particular, 3DTRL improves the baseline by a large margin in Pouring and MC, corroborating that 3DTRL adapts to diverse unseen viewpoints. The improvements of 3DTRL on Pick and MC also suggests the strong generalizability when learning from smaller datasets. With enough data (Lift & Can), 3DTRL still outperforms but the gap is small. When evaluating in **Unseen** setting, both methods have performance drop. However, 3DTRL still outperforms in Pick, MC, and Can, which suggests the representations learned by 3DTRL are able to generalize over novel viewpoints. MC has the largest viewpoint diversity so it is hard to obtain reasonable align results in the unseen setting for both the models.

Table 2: Video alignment results compared with SOTA methods. Values are alignment errors.

| Method | Backbone | Input | Pouring | Pick | MC |
|---|---|---|---|---|---|
| TCN [49] | CNN | 1 frame | 0.180 | 0.273 | 0.286 |
| Disentanglement [51] | CNN | 1 frame | - | 0.155 | 0.233 |
| mfTCN [17] | 3DCNN | 8 frames | 0.143 | - | - |
| mfTCN [17] | 3DCNN | 32 frames | 0.088 | - | - |
| DeiT [59]+TCN | Transformer | 1 frame | 0.155 | 0.216 | 0.292 |
| **+3DTRL** | Transformer | 1 frame | **0.080** | **0.116** | **0.202** |

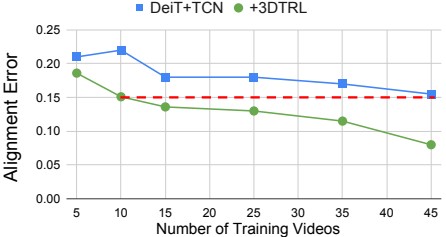

Figure 5: Alignment error w.r.t # training videos. Red dashed line indicates 3DTRL using 10 videos outperforms DeiT using 45 videos.

In Table 2, we further compare 3DTRL with previous methods on certain datasets. Note that Disentanglement [51] uses extra losses whereas others use time-contrastive loss only. We find that

3DTRL with only *single-frame* input is able to surpass the strong baselines set by models using extra losses [51] or multiple input frames [17]. We also vary the number of training videos in Pouring dataset, and results from Figure 5 show that 3DTRL benefits from more data. Meanwhile, 3DTRL can outperform the baseline while only using 22% of data the baseline used.

## 4.3 Quantitative Evaluations on Recovering 3D information

We perform quantitative evaluations on how well 3DTRL recovers 3D information. We emphasize once more that the 3D recovery in our approach is done without any supervision; it is optimized with respected to the final loss (e.g., object classification), without any access to the ground truth 3D information. We first discuss 3D estimations focusing on pseudo-depth estimation (Section 4.3.1), then we evaluate camera estimation (Section 4.3.2).

### 4.3.1 3D Estimation Evaluation

The key component of our 3D estimation is pseudo-depth estimation. In order to evaluate the 3D estimation capability, we compare the pseudo-depth map with ground truth depth map, using NeRF [40] dataset. We test the pseudo-depth with DeiT-T+3DTRL trained on IN-1K.

**Metric: Depth Correlation.** Since our estimated pseudo-depth ($d'$) and ground truth ($d$) are in different scales, we measure their relative correspondence, i.e., correlation of two sets of data. We use Pearson's r: $r = \text{correlation}(d, d')$, where we regard two depth maps as two groups of data. Note that 3DTRL operates on visual Transformers, so the pseudo-depth map is at a very coarse scale ($14 \times 14$). We also resize the given depth map to $14 \times 14$ to perform the evaluation. We report the average of $r$ across all evaluation subsets.[1]

**Results.** Figure 6 shows the evaluation results. We find the estimated pseudo-depth highly correlates with the ground truth ($r \approx 0.7$). Compared to the random prediction baseline, the depth from 3DTRL shows much higher correlation to the ground truth. We also find that the model learns the estimated depth with higher correlation in the earlier stage of the training (e.g., 20 epochs) and then drops a bit in the later training epochs ($\sim 0.07$ drop).

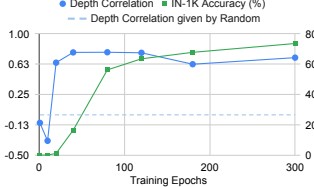 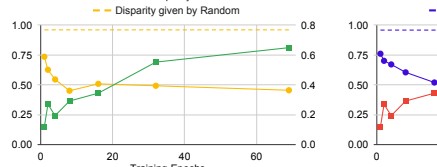 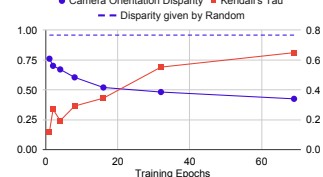

Figure 6: Depth correlation evaluation results. We find our estimation has a high ($\sim$0.7) correlation to the ground truth.

Figure 7: Quantitative evaluations on camera position (left) and camera orientation (right) respectively. Overall, we show the estimated cameras from 3DTRL have an acceptable mapping to the ground truth (both give $< 0.5$ disparity).

### 4.3.2 Camera Estimation Evaluation

In this evaluation, we mainly answer how well 3DTRL estimates camera position and orientation. To do this, we evaluate DeiT+3DTRL trained in previous video-alignment task (Section 4.2), using **Can** dataset. Recall that we train the model from scratch for the video alignment, without access to ground truth 3D information. We use first-person view videos for our evaluation — the camera moves together with the robot and our objective is to estimate its pose. We introduce two metrics.

**Metric1: Camera Position Disparity**: For each video, we get the estimated camera positions $\{\mathbf{p}'\}$ and ground truth camera positions $\{\mathbf{p}\}$, which are 3-D vectors. Since estimated and ground truth cameras are in two different coordinate systems, we need to measure how estimated camera positions map to the ground truth in a scale, translation and rotation-invariant way. The existing metrics like AUROC [30, 48] are not applicable. Therefore, we use Procrustes analysis [21] and report the disparity metric. The disparity value ranges $[0, 1]$, where a lower value indicates that two sets are more similar. We take the average disparity of all videos.

**Metric2: Camera Orientation Disparity**: Each camera has its orientation, i.e. "looking-at" direction, described by a 3-D vector. We note the estimated camera orientations $\{\mathbf{o}'\}$ and ground truth $\{\mathbf{o}\}$.

---

[1]To do so, we convert $r$ to Fisher's $z$, take the mean value across all subsets, and convert back to Pearson's $r$.

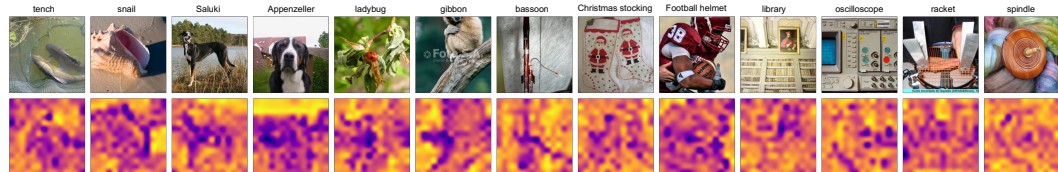

Figure 8: **Top**: IN-1K samples with corresponding class label. **Bottom**: Estimated pseudo-depth map, interpolated from $16 \times 16$ to $224 \times 224$ for better understanding. Depth increases from blueish to yellowish region. More examples are in Figure 14 in Appendix.

Similar as the camera position disparity mentioned above, we use the disparity given by Procrustes analysis [21] to measure how $\{\mathbf{o}'\}$ and $\{\mathbf{o}\}$ align. We report the mean disparity from all the videos.

**Results.**  Results are shown in Figure 7. Both position and orientation disparity of the final model are relatively small ($< 0.5$), showing that the camera estimation from 3DTRL is partially aligned with the ground truth. We also observe that the disparity decreases over the training epochs.

## 4.4  Qualitative Evaluations

We visualize some qualitative results for an intuitive understanding of 3DTRL's effectiveness. The visualization contains estimated pseudo-depth maps which are from the key intermediate step of 3DTRL. We use ImageNet-1K trained DeiT-T+3DTRL and its validation samples for visualization. In Figure 8, we observe a fine separation of the object-of-interest and the background in most of the pseudo-depth maps. Thus, these predicted pseudo-depth maps are sufficient to recover the tokens corresponding to object-of-interest in 3D space. For a qualitative evaluation on camera pose, please refer to Figure 10 in the Appendix, where we show the images with the same/similar object pose (in different environments) result in similar estimated extrinsics regardless of the background. We believe our estimated extrinsics are doing object-centric canonicalization of images with respect to their object poses, in order to optimize the representations for the final losses.

## 4.5  3DTRL on More Transformer Architectures

3DTRL is designed to be a plug-and-play model for Transformers.  We test 3DTRL with more Transformer architectures on CIFAR and two multi-view video alignment datasets. We use Swin [36] (Tiny) and TnT [24] (Small).  Results are provided in Table 3.  We find that 3DTRL generally improves the performance of two Transformer architectures in all the datasets.  This confirms 3DTRL is applicable to different Transformer architectures. The relative small improvement over the Swin backbone is due to the strong inductive bias from its local window.

Table 3: Results of using 3DTRL in more Transformer architectures, on CIFAR and multi-view video alignment datasets. Reported numbers are accuracy for CIFAR and Kendall's tau for video alignment, both are the higher the better. We show 3DTRL generally improves the performance in all tasks.

| Model | CIFAR-10 | CIFAR-100 | Pouring | Pick |
|---|---|---|---|---|
| Swin-T | 50.11 | 21.53 | 0.584 | 0.623 |
| **+3DTRL** | **50.29** (+0.18) | **21.55** (+0.02) | **0.683** (+0.099) | **0.640** (+0.017) |
| TnT-S | 81.25 | 54.07 | 0.740 | 0.640 |
| **+3DTRL** | **82.43** (+1.18) | **56.00** (+1.93) | **0.792** (+0.052) | **0.671** (+0.031) |

## 4.6  Ablation Studies

We conduct our ablation studies on image models mostly using *CIFAR* for image classification and *Pick* for multi-view video alignment.

**MLP vs. 3DTRL.**  3DTRL is implemented by several MLPs with required geometric transforms in between.  In this experiment, we replace 3DTRL with the similar number of fully-connected layers with residual connection, to have comparable parameters and computation as 3DTRL. Results are provided in Table 4. We find that MLP implementation is only comparable with the baseline performance despite the increase in parameters and computation. Thus, we confirm that the geometric transformations imposed on the token representations is the key to make 3DTRL effective.

**Token Coordinates Estimation.**  In this ablation, we show how estimating only depth compared to estimating a set of 3 coordinates $xyz$ differs in 3DTRL. We find at Line 4 and 5 in Table 4 that

Table 4: Ablation study results. For CIFAR, we test on models based on tiny (T), small (S) and base (B) backbones (DeiT) and report accuracy(%). For Pick we only test base model and report alignment error.

| Method | CIFAR-10 (T/S/B) | CIFAR-100 (T/S/B) | Pick |
|---|---|---|---|
| DeiT | 74.1/ 77.2 / 76.6 | 51.3 / 54.6 / 51.9 | 0.216 |
| DeiT + MLP | 74.2 / 77.2 / 76.5 | 47.9 / 54.7 / 53.4 | 0.130 |
| DeiT + 3DTRL | 78.8 / 80.7 / 82.8 | **53.7** / 61.5 / **61.8** | **0.116** |
| Depth Estimation → $xyz$ Estimation | 76.7 / 78.2 / 77.4 | 48.3 / 54.1 / 52.6 | 0.134 |
| Embedding → Concat. | **80.7** / **83.7** / **84.9** | 53.4 / **61.8** / 60.2 | 0.133 |

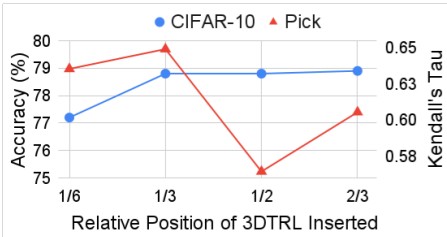

Figure 9: Results on inserting 3DTRL at different locations.

depth estimation is better because it uses precise 2D image coordinates when recovering tokens in 3D space, whereas estimating $xyz$ regresses for the 3D coordinates without any geometric constraints. Also, we find that estimating $xyz$ hampers the performance in image classification task more than video alignment. This is because estimating 3D coordinates is harder when the training samples are in unconstrained scenarios. In contrast, video pairs in an alignment dataset share the same scene captured from different view angles, which facilitates the recovery of token in 3D space.

**How to incorporate 3D positional information in Transformers?** By default, we use Equation 5 to incorporate the 3D positional information in Transformer backbone through a learned positional embedding $p^{3D}$. In this experiment, we directly infuse the estimated 3D world coordinates $p^{world}$ within the token representation by concatenating them across the channel axis. The $(m + 3)$-d feature is then projected back to $m$-d by a MLP. We keep parameters and computation comparable to the default 3DTRL. We test this 3DTRL variant (Embedding → Concat.) and results are presented in Table 4. We find that the concatenation variant outperforms the default variant of 3DTRL on CIFAR-10, but comparable and worse results in CIFAR-100 and Pick. This observation substantiates the instability of using raw 3D coordinates. In comparison, the use of 3D positional embedding is generalizable to more challenging and diverse scenarios.

**Comparison with perspective augmentation** We compare 3DTRL with vanilla data augmentation on perspective transforms on video alignment task and results are in Appendix B. We confirm 3DTRL does better than applying perspective augmentations.

**Where should we have 3DTRL?** We vary the location of 3DTRL to empirically study the optimal location of 3DTRL in a 12-layer DeiT-T backbone. In Figure 9, we find that inserting 3DTRL at the earlier layers (after 1/3 of the network) yields the best performance consistently on both datasets.

### 4.7 3DTRL for Video Representation Learning

In this section, we illustrate how 3DTRL can be adapted for video models. Since we aim at learning viewpoint-agnostic representations, as a natural choice we validate the effectiveness of 3DTRL on video datasets with multi-camera setups and cross-view evaluation. Consequently, we conduct our experiments on two multi-view action recognition datasets: **Toyota Smarthome** [12] (Smarthome) and **NTU-RGB+D** [50] (NTU) for the task of action classification. For evaluation on Smarthome, we follow Cross-View 2 (CV2) and Cross-Subject (CS) protocols proposed in [12], whereas on NTU, we follow Cross-View (CV) protocol proposed in [50]. In cross-view protocols, the model is trained on a set of cameras and tested on a different set of cameras. Similarly for cross-subject protocol, the model is trained and tested on different set of subjects. More details on these datasets are provided in Appendix I.

Table 5: Results on action recognition on Smarthome and NTU. *Acc* is classification accuracy (%) and *mPA* is mean per-class accuracy. In methods using Kinetics-400 (K400) pre-training, TimeSformer backbone is always initialized with pre-trained weights, and 3DTRL w/, w/o K400 denotes 3DTRL is randomly initialized and is initialized from pre-trained weights respectively.

| Method | Smarthome (CV2) | | Smarthome (CS) | | NTU (CV) |
|---|---|---|---|---|---|
| | *Acc* | *mPA* | *Acc* | *mPA* | *Acc* |
| TimeSformer [3] | 59.4 | 27.5 | 75.7 | 56.1 | 86.4 |
| **+ 3DTRL** | **62.9** (+3.5) | **34.0** (+6.5) | **76.1** (+0.4) | **57.0** (+0.9) | **87.9** (+1.5) |
| | Kinetics-400 pre-trained | | | | |
| TimeSformer [3] | 69.3 | 37.5 | 77.2 | 57.7 | 87.7 |
| + 3DTRL w/o K400 | 69.5 (+0.2) | 39.2 (+1.7) | 77.5 (+0.3) | 58.9 (+1.2) | **88.8** (+1.1) |
| **+ 3DTRL w/ K400** | **71.9** (+2.6) | **41.7** (+4.2) | **77.8** (+0.6) | **61.0** (+2.3) | 88.6 (+0.9) |

**Network architecture & Training / Testing**    TimeSformer [3] is a straightforward extension of ViT [16] for videos which operates on spatio-temporal tokens from videos, so that 3DTRL can be easily deployed to TimeSformer as well. Similar to our previous experimental settings, we place 3DTRL after 4 Transformer blocks in TimeSformer. Please refer to Appendix I for detailed settings.

**Results**    In Table 5, we present the action classification results on Smarthome and NTU datasets with 3DTRL plugged in TimeSformer.

3DTRL can easily take advantage of pre-trained weights because it does not change the relying backbone Transformer – just being added in between blocks. In Table 5, we present results for two fine-tuning scenarios: (a) 3DTRL w/o K400 and (b) 3DTRL w/ K400. For the first scenario (a), TimeSformer is initialized with K400 pre-training weights and leave the parameters in 3DTRL randomly initialized. Then in the fine-tuning stage, all the model parameters including those of 3DTRL is trained. In the second scenario (b), all parameters in TimeSformer and 3DTRL are pre-trained on K400 from scratch and fine-tuned on the respective datasets.

We find that all the variants of 3DTRL outperforms the baseline TimeSformer results. Our experiments show that although there is an improvement with 3DTRL compared to the baseline for different fine-tuning strategy, it is more significant when 3DTRL is pre-trained with K400. However, when large-scale training samples are available (NTU), 3DTRL does not require K400 pre-training. To sum up, 3DTRL can be seen as a crucial ingredient for learning viewpoint-agnostic video representations.

## 5   Related Work

There has been a remarkable progress in visual understanding with the shift from the use of CNNs [6, 26, 35, 58] to visual Transformers [16]. Transformers have shown substantial improvements over CNNs in image [7, 16, 24, 31, 36, 59, 65] analysis and video [1, 3, 37, 44, 47] understanding tasks due to its flexibility in learning global relations among visual tokens. Studies also combine CNNs with Transformer architectures to leverage the pros in both the structures [10, 22, 38, 52, 53]. In addition, Transformer has been shown to be effective in learning 3D representation [68]. However, these advancements in architecture-types have not addressed the issue of learning viewpoint-agnostic representation. Viewpoint-agnostic representation learning is drawing increasing attention in the vision community due to its wide range of downstream applications like 3D object-detection [46],video alignment [8, 18, 19], action recognition [54, 55], pose estimation [25, 57], robot learning [4, 27, 29, 49, 51, 56], and other tasks.

There is a broad line of work towards directly utilizing 3D information like depth [25], pose [13, 14], and point clouds [43, 45], or in some cases deriving 3D structure from paired 2D inputs [61]. However, methods rely on the availability of multi-modal data which is hard to acquire are not scalable.

Consequently, other studies have focused on learning 3D perception of the input visual signal in order to generalize the learned representation to novel viewpoints. This is done by imposing explicit geometric transform operations in CNNs [5, 28, 42, 46, 64], without the requirement of any 3D supervision. In contrast to these existing works, our Transformer-based 3DTRL imposes geometric transformations on visual tokens to recover their representation in a 3D space. To the best of our knowledge, 3DTRL is the first of its kind to learn a 3D positional embedding associated with the visual tokens for viewpoint-agnostic representation learning in different image and video tasks.

## 6   Conclusion

In this work, we have presented 3DTRL, a plug-and-play module for visual Transformer that leverages 3D geometric information to learn viewpoint-agnostic representations. Within 3DTRL, by pseudo-depth estimation and learned camera parameters, it manages to recover positional information of tokens in a 3D space. Through our extensive experiment, we confirm 3DTRL is generally effective in a variety of visual understanding tasks including image classification, multi-view video alignment, and cross-view action recognition, by adding minimum parameters and computation overhead.

## Acknowledgment

We thank insightful discussions with members of Robotics Lab at Stony Brook. This work is supported by Institute of Information & communications Technology Planning & Evaluation (IITP) grant funded by the Ministry of Science and ICT (No.2018-0-00205, Development of Core Technology of Robot Task-Intelligence for Improvement of Labor Condition. This work is also supported by the National Science Foundation (IIS-2104404 and CNS-2104416).

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
