## A    Qualitative Evaluation on Camera Estimation

We collected images of the same object with various object poses and backgrounds, and visualized their camera estimation in Figure 10. This was done with DeiT-T+3DTRL trained on ImageNet. We observe 3DTRL estimates similar camera poses (clustered in the red-dashed circle) for similar object poses regardless of different backgrounds. When given different object poses, 3DTRL estimates cameras in scattered position and orientations. We notice that there is an outlier whose estimation is also introduced in this cluster, owing to the model invariance to horizontal flipping which is used as an augmentation during training. This visualization suggests that 3DTRL, when trained for the object classification task, might be performing object-centric canonicalization of the input images.

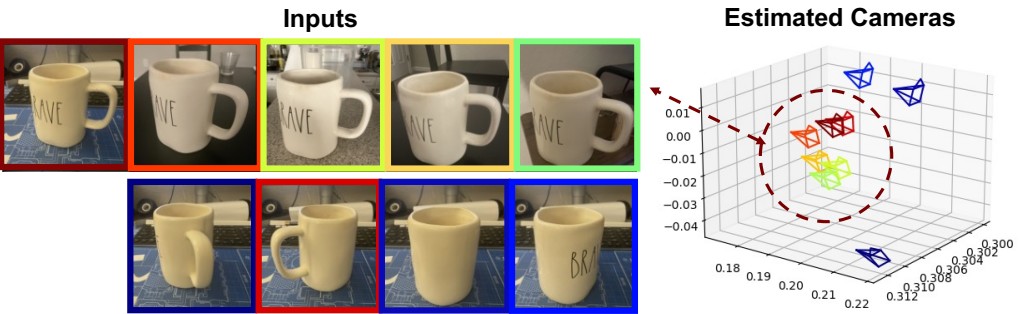

Figure 10: Qualitative experiment on how 3DTRL reacts to similar and different poses of the same object. Photos are taken by a regular smartphone. We use DeiT-T+3DTRL trained on ImageNet. The estimated cameras from the similar poses (first row) are clustered as shown in the red-dashed circle, while the other cameras from different poses (second row) are apart. We notice an outlier whose estimation is also introduced in this cluster, owing to the model invariance to horizontal flipping which is used as an augmentation during training.

We also present more qualitative visualizations of estimated camera positions (Figure 11) and estimated 3D world locations of image patches (Figure 12). We find that the estimations approximately reflect the ground truth or human perception which the model has no access to during training. These estimations are not necessarily required to be perfectly aligned with ground truth, but the results show that they are reasonable and sufficient for providing 3D information.

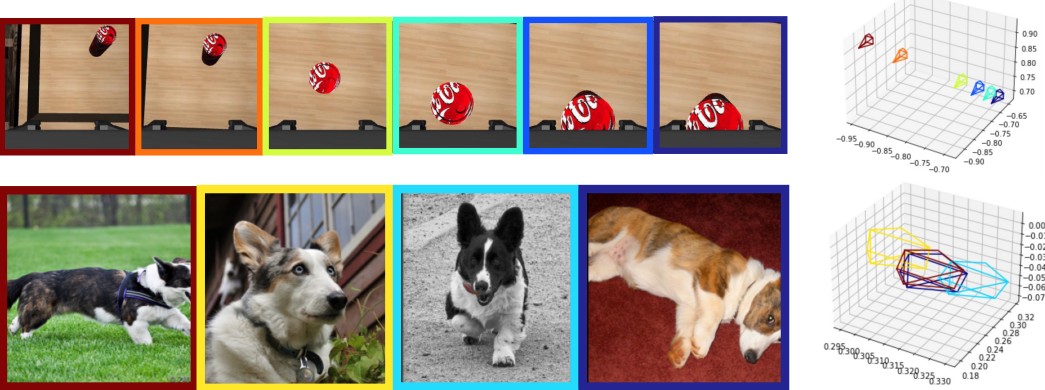

Figure 11: Visualization of image samples (left, with colored boundaries) and estimated camera positions in a 3D space (right). The color of the boundary on each image corresponds to the estimated camera from that image. **Top**: Images are from a video clip captured by an egocentric (eye-in-hand) camera on a robot arm in Can environment, ordered by timestep from left to right. The estimated camera positions approximately reflects the motion of the robot arm, which is moving towards right and down. **Bottom**: Samples from ImageNet-1K. The estimated camera pose of the second image (yellow boundary) is somehow at a head-up view, and the rest are at a top-down view. These estimated cameras are approximately aligned with human perception.

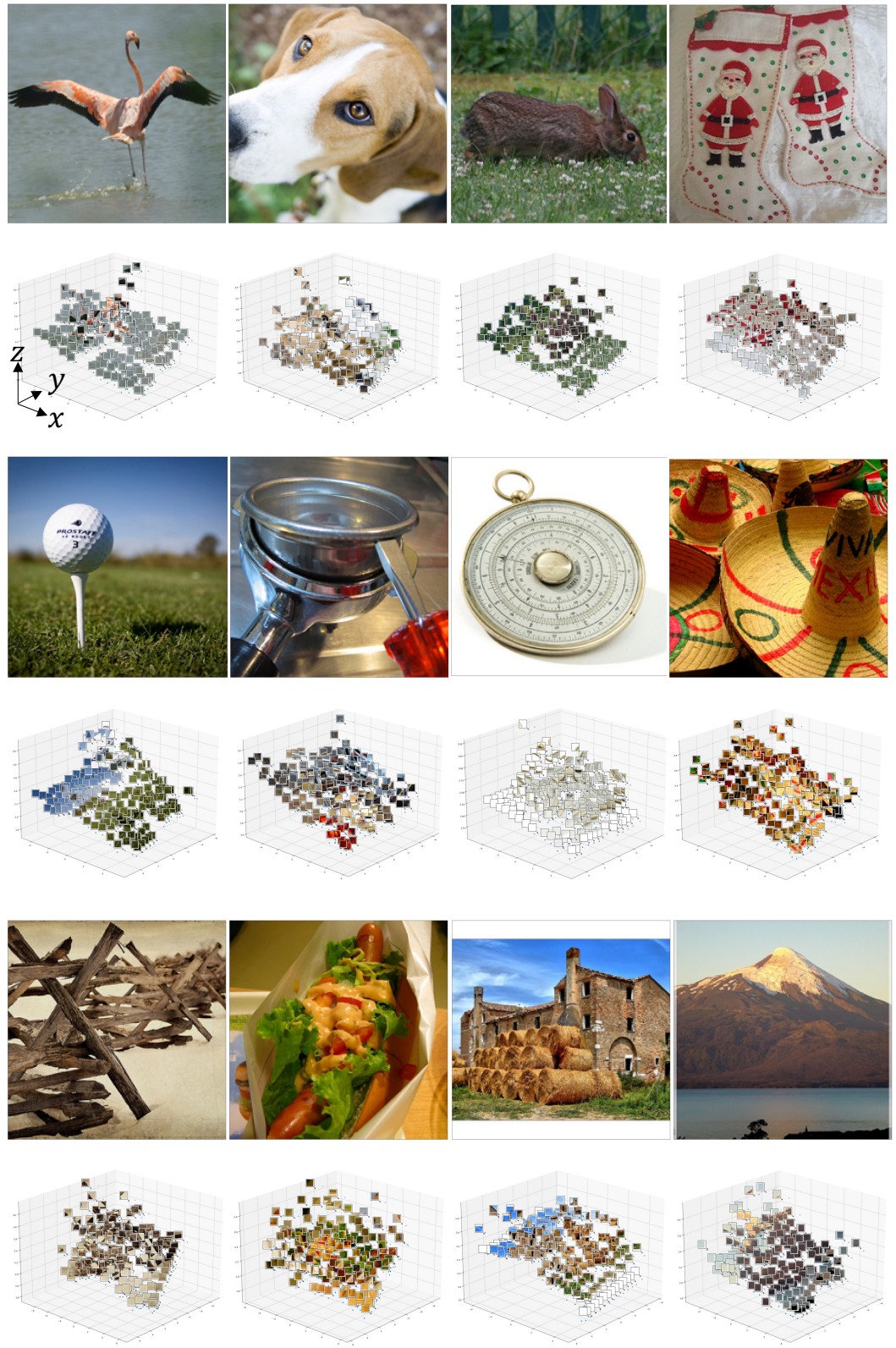

Figure 12: Visualization of image patches at their estimated 3D world locations. The center of the $xy$-plane (horizontal plane, at the bottom) is the origin of the 3D space. The vertical axis is $z$-axis. The patches corresponds to object-of-interest usually have larger $z$ values (corresponding to larger pseudo-depth values), which are localized "farther" from the origin. Most of the background patches have smaller $z$ values that are at the "closer" to the $xy$-plane.

# B Experiment on Perspective Augmentation

In this experiment, we investigate whether perspective augmentation applied to input images will help the model to learn viewpoint-agnostic representations. We test with DeiT+3DTRL on multi-view video alignment task. The training procedure is the same for all variants. Results are shown in Table 6. From the results we show that naively adding perspective augmentation does not improve the viewpoint-agnostic representation learning. Instead, it harms the performance compared to the DeiT baseline, since the perspective augmentation is overly artificial compared to the real-world viewpoint changes. Such augmentation does not contribute to viewpoint-agnostic representation learning.

Table 6: Comparison between 3DTRL and perspective augmentation on training data. Overall, perspective augmentation shows a negative effect on all the tasks, because the perspective augmentation on image is not the real viewpoint change.

| Model | Pouring | Pick | MC | Can | Lift |
|---|---|---|---|---|---|
| DeiT | 0.426 | 0.244 | -0.115 | 0.789 | 0.716 |
| DeiT + Perspective Augmentation | 0.200 | -0.249 | -0.419 | 0.342 | 0.486 |
| DeiT + 3DTRL | **0.740** | **0.635** | **0.392** | **0.824** | **0.739** |

# C Discussion on Pseudo-depth Estimation

Most of images from ImageNet have a simple scene (background), so it's easier for the pseudo-depth estimation to focus on objects. In examples shown in Figure 13, we show that the pseudo-depth is also estimated for other foreground objects apart from the primary class object.

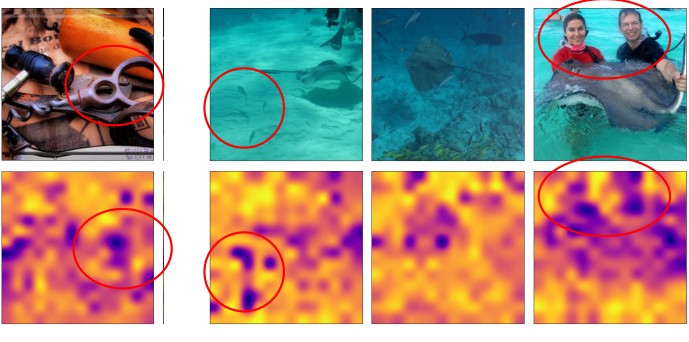

Label: Hammer          Label: Stingray

Figure 13: Examples of pseudo-depth estimation, applied to images with multiple types of objects. We use red circles to highlight the locations of non-class objects. In the hammer input, depth is also estimated on the other tool. In the first sample of stingray, three other fishes in the "foreground" corresponds to a lower depth value. In the last sample of stingray, two humans are predicted with a relative lower depth value, in between the stingray and the background.

# D More Ablation Studies

## D.1 How many 3DTRLs should be used?

We explore the possibility of using multiple 3DTRLs in a Transformer backbone. This usage is essentially estimating 3D information from different levels of features, and injecting such information back into the backbone. We test several DeiT-T variants using multiple 3DTRLs on CIFAR-10 and results are shown in Table 7. We find that using multiple 3DTRLs further increases the performance in general compared to using only one 3DTRL. This shows the extra capacity from multiple 3DTRLs benefits representation learning. Specifically, we demonstrate that inserting 3DTRLs at layer 4, 6,

and 8 yields the best result among all the strategies we explore. This experiment empirically shows multiple 3DTRLs potentially benefit the model.

Table 7: CIFAR-10 Performance when using multiple 3DTRLs based on DeiT-T.

| 3DTRL Location(s) | N/A (DeiT baseline) | 4 | 4, 6, 8 | 4, 4, 4 | 2, 4, 6, 8 |
|---|---|---|---|---|---|
| CIFAR-10 | 74.1 | 78.8 | **79.5** | 79.3 | 79.1 |

## D.2 Regularization effect of 3DTRL

Mixup [67] and CutMix [66] are commonly used image augmentation methods in training image models, which mixes two images to create an augmented image for training input. Such technique along with other augmentations provides diverse image samples so that the image model is regularized and avoids overfitting. We hypothesise that Mixup & CutMix could potentially damage structures in original image, which may cause inaccurate depth estimation in 3DTRL, thus hamper the training procedure. Therefore, we conduct an experiment on ImageNet-1K of disabling Mixup & CutMix. We train baseline DeiT and DeiT+3DTRL from scratch, and compare the results in both original validation set and perturbed set. In Table 8, we find that both baseline and our method increases validation scores after disabling Mixup & CutMix, and 3DTRL still outperforms the baseline by 0.2%. However, when tested on view-perturbed set, the baseline model shows a great performance drop (-6.8%) which is much larger than 3DTRL (-1.6%).

Table 8: ImageNet-1K and ImageNet-1K-Perturbed results when Mixup & Cutmix are disabled.

| Model | ImageNet-1K | ImageNet-1K-Perturbed |
|---|---|---|
| DeiT-T | 73.4 | 61.3 |
| DeiT-T + Mixup & CutMix Disabled | 74.2 | 54.5 |
| DeiT-T+3DTRL | 73.6 | 64.6 |
| DeiT-T+3DTRL + Mixup & CutMix Disabled | 74.4 | 63.0 |

## D.3 Camera parameter estimation in video model

We implemented and evaluated two different strategies for the camera parameter estimation $g(\cdot)$ in 3DTRL for videos. These two strategies are: (a) Divided-Temporal (DT) and (b) Joint-Temporal (JT) estimation introduced below. In (a) DT strategy, we estimate one set of camera parameters $[\mathbf{R}|\mathbf{t}]_t$ per input frame ($S_t$) in a dissociated manner, and thus estimate a total of $T$ camera matrices for the entire video. In (b) JT strategy, we estimate only one camera from all frames $S = \{S_1, \ldots, S_T\}$. The camera is shared across all spatial-temporal tokens associated with the latent 3D space. The underlying hypothesis is that the camera pose and location do not change during the video clip. JT could be helpful to properly constrain the model for scenarios where camera movement is not required, but it is not generalizable to scenarios where the subject of interest quite often moves within the field-of-view.

By default, we used DT strategy in all experiments presented in the main paper. In Table 9, we show a comparison of DT and JT strategies in different scenarios. Note that 3DTRL implemented with JT strategy under-performs the baseline TimeSformer on Smarthome (in most of the CV2 experiments) dataset. We find that the JT strategy adoption for video models is particularly effective when there is a large availability of training data, for example on NTU dataset. However, these results with JT strategy are inconsistent across different datasets and also less substantial w.r.t. the results with our default DT strategy. This shows the requirement of estimating camera matrix per frame rather than a global camera matrix for video representation tasks.

# E   Limitations

We find that 3DTRL suffers when estimating small objects in the scene, or estimating objects in a complex scene, due to the coarse scale (in 16x16 image patches) from the backbone Transformer. One

Table 9: Comparison of DT and JT strategies in 3DTRL for action recognition task.

| Method | Strategy | Smarthome (CV2) | | Smarthome (CS) | | NTU (CV) |
| --- | --- | --- | --- | --- | --- | --- |
| | | *Acc* | *mPA* | *Acc* | *mPA* | *Acc* |
| TimeSformer [3] | - | 59.4 | 27.5 | 75.7 | 56.1 | 86.4 |
| **+ 3DTRL** | DT | **62.9** (+3.5) | **34.0** (+6.5) | 76.1 (+0.4) | 57.0 (+0.9) | **87.9** (+1.5) |
| + 3DTRL | JT | 58.6 (-0.8) | 30.9 (+3.4) | **76.2** (+0.5) | **57.2** (+1.1) | **87.9** (+1.5) |
| **Kinetics-400 pre-trained** | | | | | | |
| TimeSformer [3] | - | 69.3 | 37.5 | 77.2 | 57.7 | 87.7 |
| + 3DTRL w/o K400 | DT | 69.5 (+0.2) | 39.2 (+1.7) | 77.5 (+0.3) | 58.9 (+1.2) | **88.8** (+1.1) |
| **+ 3DTRL w/ K400** | DT | **71.9** (+2.6) | **41.7** (+4.2) | **77.8** (+0.6) | **61.0** (+2.3) | 88.6 (+0.9) |
| + 3DTRL w/o K400 | JT | 66.6 (-2.7) | 35.0 (-2.5) | 77.0 (-0.2) | 58.6 (+0.9) | 88.6 (+0.9) |
| + 3DTRL w/ K400 | JT | 68.2 (-0.9) | 37.1 (-0.4) | 77.0 (-0.2) | 59.9 (+2.2) | 87.7 (+0.0) |

possible solution is to decrease the patch size or enlarge the input size in the backbone Transformer, but in practice it is computationally infeasible as Attention complexity grows quadratically. Similar problem occurs when 3DTRL is applied on Transformers having hierarchical architectures like Swin Transformer [36], where we find our improvement is minor compared to DeiT. In hierarchical architectures, image patches are merged after one stage so the resolution of the pseudo-depth map decreases quadratically. To solve this issue, we recommend to place 3DTRL at the location before any patch merging in such hierarchical Transformers.

# F  Implementation Details

3DTRL is easily inserted in Transformers. The components of 3DTRL are implemented by several MLPs and required geometric transformations in between. We keep the hidden dimension size in MLPs the same as the embedding dimensionality of Transformer backbone, Tiny=192, Small=384, Base=768 in specific. We provide PyTorch-style pseudo-code about inserting 3DTRL in Transformer (Algorithm 1) and about details of 3DTRL (Algorithm 2). We use image Transformer for example and omit operations on *CLS* token for simplicity. Full implementation including video model is provided in supplementary files.

---

**Algorithm 1:** PyTorch-style pseudo-code for using 3DTRL in Transformer

```
# Use 3DTRL with Transformer backbone
Class Transformer_with_3DTRL:
  def __init__(self, config):
    # Initialize a Transformer backbone and 3DTRL
    self.backbone = Transformer(config)
    self.3dtrl = 3DTRL(config)
    # Before which Transformer layer we insert 3DTRL
    self.3dtrl_location = config.3dtrl_location

  def forward(self, tokens):
    for i, block in enumerate(self.backbone.blocks):
      # Tokens go through 3DTRL at desired insert location
      if i == self.3dtrl_location:
        tokens = self.3dtrl(tokens)
      # Tokens go through backbone layers
      tokens = block(tokens)
    return tokens
```

---

**Algorithm 2:** PyTorch-style pseudo-code for 3DTRL

```python
Class 3DTRL:
    # Make a 3DTRL
    def __init__(self, config):
        # 2D coordinates on image plane
        self.u, self.v = make_2d_coordinates()
        # Depth estimator
        self.depth_estimator = nn.Sequential(
            nn.Linear(config.embed_dim, config.embed_dim),
            nn.ReLU(),
            nn.Linear(config.embed_dim, 1))

        # Camera parameter estimator, including a stem and two heads
        self.camera_estimator_stem = nn.Sequential(
            nn.Linear(config.embed_dim, config.embed_dim),
            nn.ReLU(),
            nn.Linear(config.embed_dim, config.embed_dim),
            nn.ReLU(),
            nn.Linear(config.embed_dim, 32),
            nn.ReLU(),
            nn.Linear(32, 32))
        # Heads for rotation and translation matrices.
        self.rotation_head = nn.Linear(32, 3)
        self.translation_head = nn.Linear(32, 3)

        # 3D positional embedding layer
        self.3d_pos_embedding = nn.Sequential(
            nn.Linear(3, config.embed_dim),
            nn.ReLU(),
            nn.Linear(config.embed_dim, config.embed_dim))

    def forward(self, tokens):
        # Depth estimation
        depth = self.depth_estimator(tokens)
        camera_centered_coords = uvd_to_xyz(self.u, self.v, depth)

        # Camera estimation
        interm_rep = self.camera_estimator_stem(tokens)
        rot, trans = self.rotation_head(interm_rep),
         self.translation_head(interm_rep)
        rot = make_rotation_matrix(rot)

        # Transformation from camera-centered to world space
        world_coords = transform(camera_centered_coords, rot, trans)

        # Convert world coordinates to 3D positional embeddings
        3d_pos_embed = self.3d_pos_embedding(world_coords)

        # Generate output tokens
        return tokens + 3d_pos_embed
```

## G    Settings for Image Classification

**Datasets**    For the task of image classification, we provide a thorough evaluation on three popular image datasets: CIFAR-10 [34], CIFAR-100 [34], and ImageNet [15]. CIFAR-10/100 consists of 50k training and 10k test images, and ImageNet has 1.3M training and 50k validation images.

**Training Configurations**    We follow the configurations introduced in DeiT [59]. We provide a copy of configurations here in Table 10 (CIFAR) and Table 11 (ImageNet-1K) for reference. We use 4 NVIDIA Tesla V100s to train models with Tiny, Small and Base backbones on ImageNet-1K for ∼22 hours, ∼3 days and ∼5 days respectively.

Table 10: CIFAR Training Settings

| | |
|---|---|
| Input Size | 32×32 |
| Patch Size | 2×2 |
| Batch Size | 128 |
| Optimizer | AdamW |
| Optimizer Epsilon | 1.0e-06 |
| Momentum | $\beta_1, \beta_2 = 0.9, 0.999$ |
| layer-wise lr decay | 0.75 |
| Weight Decay | 0.05 |
| Gradient Clip | None |
| Learning Rate Schedule | Cosine |
| Learning Rate | 1e-3 |
| Warmup LR | 1.0e-6 |
| Min LR | 1e-6 |
| Epochs | 50 |
| Warmup Epochs | 5 |
| Decay Rate | 0.988 |
| drop path | 0.1 |
| Exponential Moving Average (EMA) | True |
| EMA Decay | 0.9999 |
| Random Resize & Crop Scale and Ratio | (0.08, 1.0), (0.67, 1.5) |
| Random Flip | Horizontal 0.5; Vertical 0.0 |
| Color Jittering | None |
| Auto-agumentation | rand-m15-n2-mstd1.0-inc1 |
| Mixup | True |
| Cutmix | True |
| Mixup, Cutmix Probability | 0.8, 1.0 |
| Mixup Mode | Batch |
| Label Smoothing | 0.1 |

## H    Settings for Video Alignment

**Datasets**    We provide the statistics of 5 datasets used for video alignment in Table 12. In general, datasets with fewer training videos, more/diverse viewpoints, and longer videos are harder for alignment. We will also provide the copy of used/converted dataset upon publish.

**Training Configurations**    The training setting for video alignment is listed in Table 13. The setting is the same for all datasets and all methods for fair comparison. GPU hours required for training vary across datasets, depending on the size of datasets and early stopping (convergence). Approximately we use 24 hours in total to fully train on all 5 datasets using an NVIDIA RTX A5000.

Table 11: ImageNet-1K Training Settings [59]

| | |
|---|---|
| Input Size | 224×224 |
| Crop Ratio | 0.9 |
| Batch Size | 512 |
| Optimizer | AdamW |
| Optimizer Epsilon | 1.0e-06 |
| Momentum | 0.9 |
| Weight Decay | 0.3 |
| Gradient Clip | 1.0 |
| Learning Rate Schedule | Cosine |
| Learning Rate | 1.5e-3 |
| Warmup LR | 1.0e-6 |
| Min LR | 1.0e-5 |
| Epochs | 300 |
| Decay Epochs | 1.0 |
| Warmup Epochs | 15 |
| Cooldown Epochs | 10 |
| Patience Epochs | 10 |
| Decay Rate | 0.988 |
| Exponential Moving Average (EMA) | True |
| EMA Decay | 0.99992 |
| Random Resize & Crop Scale and Ratio | (0.08, 1.0), (0.67, 1.5) |
| Random Flip | Horizontal 0.5; Vertical 0.0 |
| Color Jittering | 0.4 |
| Auto-agumentation | rand-m15-n2-mstd1.0-inc1 |
| Mixup | True |
| Cutmix | True |
| Mixup, Cutmix Probability | 0.5, 0.5 |
| Mixup Mode | Batch |
| Label Smoothing | 0.1 |

Table 12: Statistics of multi-view datasets used for video alignment.

| Dataset | # Training/Validation/Test Videos | # Viewpoints | Average Frames/Video |
|---|---|---|---|
| Pouring | 45 / 10 / 14 | 2 | 266 |
| MC | 4 / 2 / 2 | 9 | 66 |
| Pick | 10 / 5 / 5 | 10 | 60 |
| Can | 200 / 50 / 50 | 5 | 38 |
| Lift | 200 / 50 / 50 | 5 | 20 |

Table 13: Training Settings for Video Alignment

| | |
|---|---|
| Positive Window of TCN Loss | 3 frames in MC, Pick, Pouring; 2 frames in Can and Lift |
| Learning Rate | 1e-6 |
| Batch Size | 1 |
| Optimizer | Adam |
| Gradient Clip | 10.0 |
| Early Stopping | 10 epochs |
| Random Seed | 42 |
| Augmentations | No |

# I  Settings for Video Representation Learning

**Datasets**  Our dataset choices are based on multi-camera setups in order to provide cross-view evaluation. Therefore, we evaluate the effectiveness of 3DTRLon two multi-view datasets Toyota Smarthome [12] and NTU-RGB+D [50]. We also use Kinetics-400 [32] for pre-training the video backbone before plugging-in 3DTRL.

Toyota-Smarthome (Smarthome) is a recent ADL dataset recorded in an apartment where 18 older subjects carry out tasks of daily living during a day. The dataset contains 16.1k video clips, 7 different camera views and 31 complex activities performed in a natural way without strong prior instructions. For evaluation on this dataset, we follow cross-subject ($CS$) and cross-view ($CV_2$) protocols proposed in [12]. We ignore protocol $CV_1$ due to limited training samples.

NTU RGB+D (NTU) is acquired with a Kinect v2 camera and consists of 56880 video samples with 60 activity classes. The activities were performed by 40 subjects and recorded from 80 viewpoints. For each frame, the dataset provides RGB, depth and a 25-joint skeleton of each subject in the frame. For evaluation, we follow the two protocols proposed in [50]: cross-subject (CS) and cross-view (CV).

Kinetics-400 (K400) is a large-scale dataset with 240k training, 20k validation and 35k testing videos in 400 human action categories. However, this dataset do not posses the viewpoint challenges, we are addressing in this paper. So, we use this dataset only for pre-training purpose as used by earlier studies.

**Training Configurations**  We use clips of size $8 \times 224 \times 224 \times 3$, with frames sampled at a rate of 1/32. We use a ViT-B encoder with patch size $16 \times 16$. The training setting for action recognition on both datasets follow the configurations provided in [3]. We train all the video models on 4 RTX 8000 GPUs with a batch size of 4 per GPU for 15 epochs. A gradient accumulation is performed to have an effective batch size of 64. Similar to [3], we train our video models with SGD optimiser with 0.9 momentum and $1e - 4$ weight decay. During inference, we sample one and 10 temporal clips from the entire video on NTU and Smarthome datasets respectively. We use 3 spatial crops (top-left, center, bottom-right) from each temporal clip and obtain the final prediction by averaging the scores for all the crops.

# J  More Pseudo-depth Estimation Visualization

Figure 14 gives examples of more pseudo-depth maps.

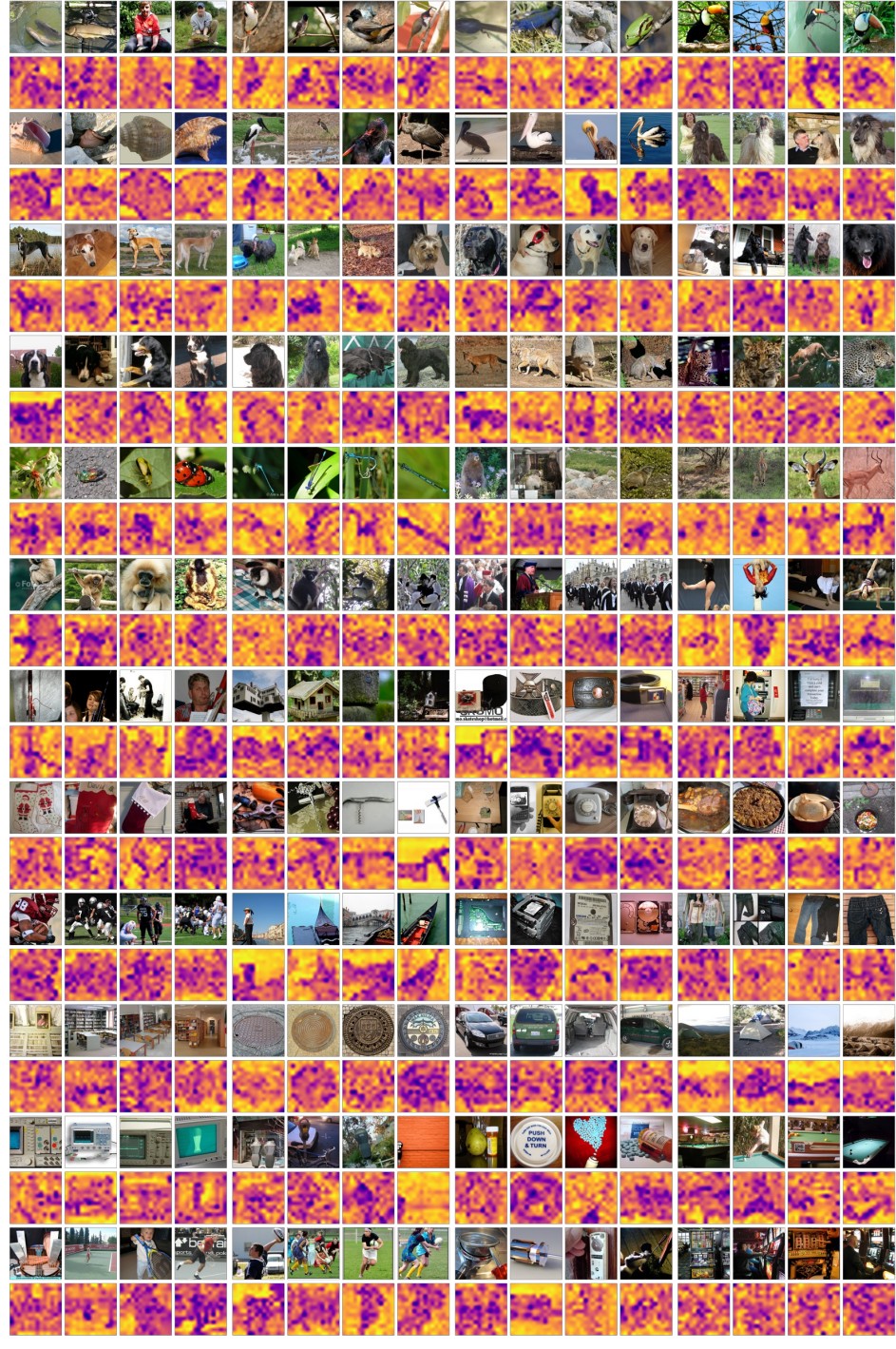

Figure 14: More examples of pseudo-depth maps.

## K    Pseudo-depth Map Visualization over Training Epochs

Figure 15 gives examples of pseudo-depth estimation over Training Epochs. We note that the results are from training 3DTRL with IN-1K. We find that the estimation varies significantly from epoch 10 to epoch 40 (higher foreground-background correctness, less missing parts of objects), but changes only a bit from epoch 40 to epoch 200 and finally to epoch 300 (mostly scales). This observation is also coherent with our quantitative evaluation in Section 4.3.1. Thus, the pseudo-depth estimation learns promptly, however the model convergence takes longer time since we are optimizing for a downstream task (eg. classification).

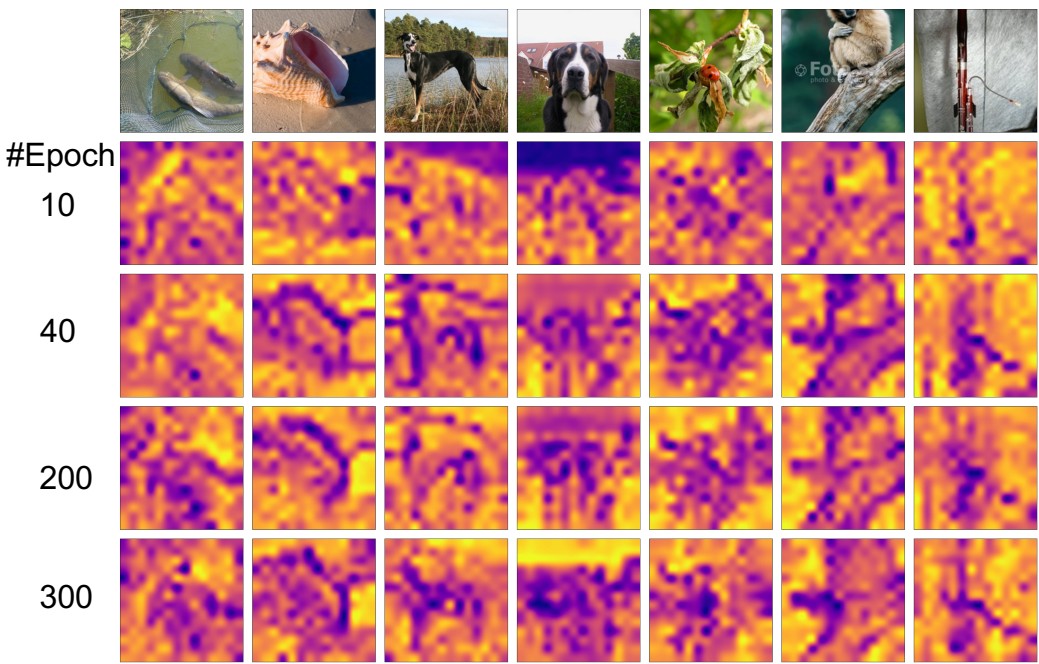

Figure 15: Pseudo-depth maps over training epochs.