# OpenReview forum: "Learning Viewpoint-Agnostic Visual Representations by Recovering Tokens in 3D Space"
_NeurIPS.cc/2022/Conference — NeurIPS 2022 Accept_

### Official Review · Reviewer_kBS4 · 2022-07-09

**Rating:** 5
**Confidence:** 4
**Soundness:** 3 good
**Presentation:** 4 excellent
**Contribution:** 3 good

**Summary:**

This work presents 3DTRL, a plug-and-play module aiming to learn viewpoint-agnostic representations by incorporating 3D geometric constraints in the learning process. The module predicts both the (pseudo) depth estimate and the camera parameters such that it can project the learned tokens into 3D world coordinates. The authors have shown that the module, when used together with visual Transformers, can achieve better performance in a range of visual understanding tasks, including image classification, multi-view video alignment, and cross-view action recognition.

**Questions:**

Please see the questions in the weaknesses section.

**Limitations:**

The authors did not explicitly discuss the limitations of the proposed method, but more discussions/evaluations about the model's ability to infer 3D would be needed.

I do not see any potential negative societal impact from this work.

**Strengths And Weaknesses:**

[Strengths]

I like the idea of incorporating 3D geometric information into the learning pipeline. The added module is simple to use and only casts negligible overhead regarding the parameter count and computation time but is seemingly effective in boosting the performance of existing Transformer architectures.

The authors have conducted extensive evaluations on a range of tasks involving images and videos. The proposed module is effective in all evaluated tasks and has consistently improved over the existing baselines. The authors have also shown a range of ablation studies justifying the design of the added module regarding the architecture, the type of 3D information to infer, and the way to incorporate 3D geometric constraints. These are all important lessons that can be helpful for the readers to have a deeper understanding of the method.


[Weaknesses]

The primary concern I have about this paper is that I do not fully understand how well the added module can infer the 3D information.

For example, the proposed module predicts the depth information and the camera parameters from a single camera, which will for sure have a lot of ambiguities (i.e., different combinations of depth and camera parameters could explain the same visual observation). It is unclear how the module resolves the ambiguities and how such ambiguities factor in the final performance.

I'm also not convinced about the module's 3D inference capability. It is great that the authors show the 3D prediction results in Figure 8 of the supplementary materials. However, the results do not seem to be very satisfying. I know that this is a very hard task, but no matter the size of the object, the distance from the camera to the object, and the distance between the foreground and the background, the 3D prediction results all seem similar without much differences from each other. If the predicted 3D information does not reasonably reflect the underlying 3D contents, it is a bit hard for me to justify its benefit to the downstream task performance.

Correct me if I am wrong, but the authors seem to predict only the camera extrinsic parameters but ignore the intrinsic parameters. However, the dataset will for sure contain images taken by cameras with different intrinsic parameters, which will greatly impact how the 3D points in the camera frame are projected to the 2D image plane; thus, I'm curious about how much the negligence of the intrinsic parameters influences the final performance.

The authors have also ablated the model by using different numbers of 3DTRLs in the architecture and have shown improved performance when using multiple modules. It would be thus quite interesting to see how the inferred 3D information changes as the model use different numbers of 3DTRLs.

I'm also not entirely sure how the authors specify the origin of the world coordinates; Or is the origin purely learned in an unconstrained and unsupervised way (i.e., whatever coordinates produced by the neural network)?

Due to the reasons above, I strongly suggest the authors show results in scenarios where we have access to the ground truth 3D information (e.g., simulation or some 3D scanned scenes) and systematically evaluate how well the proposed module can recover the underlying 3D contents.


[Minor]

Line 222: Figure 3 should be Table 3, but however, Table 3 currently is a figure. There might be some inconsistencies in the LaTeX code.

---

> ### Author Response · Authors · 2022-08-02
> **Authors' Response to Reviewer kBS4**
>
> We thank the reviewer for the encouragement and the insightful comments.
>
> - > Comparison against the ground truth
>   * We thank the reviewer for the suggestion. Following this, we conducted an additional experiment quantitatively evaluating the 3D information accuracy (**Section A** in the updated supplementary material). We used the dataset with the ground truth 3D information, and measured the correlation between the ground truth and our estimates. We confirm high correlation between them.
>
> - > Ambiguities in estimation
>   * We agree with the reviewer that there is scale ambiguity. As our method does not utilize any supervised 3D information, instead of directly resolving the ambiguity, we optimize the entire model including 3DTRL with respect to the downstream task (e.g., object classification). The intuition is that 3DTRL has to learn to set the pseudo-depth (and the corresponding camera parameters) at the right scale, in order to make the entire task (e.g., object recognition) successful. The resulting 3D estimation will not be metric (as it is not explicitly optimized to recover the ground truth 3D) but have its own scale, and we find that this is ok as long as it is consistent across different images.
>
> - > 3D inference capability
>   * We conducted an additional experiment to explicitly evaluate the correspondence between the estimated pseudo-depth and the ground truth depth (**Section A.1** in the updated supplementary material), and confirmed that they are correlated -- the relative relations between the estimated 3D positions match with those of the ground truth. This was reported in terms of Person’s r. Overall our estimation gives $r=0.7$ which shows a good correlation. We believe such 3D information is sufficient to contribute meaningfully as a positional embedding.
>
> - > Intrinsics
>   * Our work focuses on the approximation. We tried different intrinsics and it did not make much difference.
>
> - > Origin
>   * We set the origin to be $(0,0,0)$, and we can interpret as the camera poses with respect to this origin are implicitly learned in an unsupervised way. In **Figure 10** in the updated supplementary, we show the same object in similar object poses are predicted to have similar camera poses. We believe our estimated extrinsics are doing object-centric canonicalization of images with respect to their object poses. This aligns with the Reviewer **xnSs**’s insight: "I think we expect the poses (extrinsics) are doing some canonicalization of the input imagery -- registering them closer to a common pose."
>
> - > Different number of 3DTRLs
>   * We did not find a conclusion on how the estimation changes across multiple 3DTRLs because this kind of usage is not our main study, considering the improvement from using multiple 3DTRLs is less than the improvement we get from baseline->one 3DTRL, but the parameter and computation overhead are doubled/tripled.

---

> > ### Comment · Reviewer_kBS4 · 2022-08-08
> > **Thank you for your response**
> >
> > Thank you for the detailed response and the additional experimental results, which address the majority of my concerns. Although I'm not entirely satisfied with the 3D prediction results, I acknowledge that this is a very hard task of inferring the 3D in an unsupervised way. Given the experimental results that show clear improvements in the downstream task and the additional more detailed analysis, I'm willing to increase my score to 5.

---

> ### Author Response · Authors · 2022-08-08
> **We are looking forward to hearing from you**
>
> We thank you for the review. We tried our best to respond to the comments, including the new experiments on comparing the estimated 3D information against ground truth. It will be great if you can please check it and let us know whether we have addressed your concerns. If there is something still missing, we would be happy to add it.

---

### Official Review · Reviewer_6XtT · 2022-07-10

**Rating:** 7
**Confidence:** 4
**Soundness:** 3 good
**Presentation:** 3 good
**Contribution:** 3 good

**Summary:**

In this paper, the authors claim that deep learning-based models applied to 2D images lack 3D information that is easily understood by human vision. In turn, the authors focus on transformer-based architectures. The authors propose a 3D Token Representation Layer (3DTRL), a plug-and-play module for transformer-based architectures to learn viewpoint-agnostic representation for the visual data. Authors do this by lifting 2D token locations to 3D by a pseudo depth estimator, and a pseudo camera parameter estimator. Through experiments on Image Classification datasets and multi-view video alignment datasets, authors show the effectiveness of 3DTRL modules being robust to viewpoint changes.


**Questions:**

How did you come up with eq (3)? This is not a standard pinhole model
How would 3DTRL compare with doing some perspective augmentation while training? Isn’t that also learning viewpoint agnostic representations?
Did you try 3DTRL with the new transformer architectures such as Swin [3]? Does 3DTRL still help with such architectures?


**Limitations:**

The authors have not mentioned any limitations or negative social impact of the work. The clear limitation of the 3DTRL module I see is it is not generic module, and mostly applicable for data with multiple views of the same scene. (i.e. multiview video alignment)


**Strengths And Weaknesses:**

Strengths:

1. Authors' motivation for a need to embed 3D information in tokens is clear
2. Proposed 3DTRL is a plug-and-play module, that can be embedded into any transformer architecture
3. Authors show experiments on multiple datasets to show the effectiveness of the module

Weakness:

While the authors' motivation for the need to embed 3D information into learning visual representation is clear, the proposed approach is not convincing for multiple reasons,
1. Authors have proposed a pseudo depth estimator and pseudo camera parameter estimator to lift 2D token locations to 3D. While I can understand 2D image information being used to learn depth, in a free-view image analysis scenario, I don’t think camera parameter estimation makes sense. I.e. We don't even know the reference, and how the R,t of one image is related to the R, t of the other image in CIFAR-10? In 3D multi-view datasets there are approaches that directly regress camera poses and scene coordinates ([1], [2]). However, in this case, the authors are trying to embed 3D scene information in model weights, while in free-viewpoint images that don't have a common reference frame, the proposed approach is most likely not valid.

2. The transformers are effective with large amounts of data, and the minor improvement of 3DTRL on the ImageNet-1K dataset shows that transformers already become effective without 3DTRL. I understand that authors have shown a validation set on view-point perturbed data with the perspective transformation of the validation-set of ImageNet-1K, but this is not a valid viewpoint change, and we cannot trust that this will really work in multi-view scenarios.

3. While  I can understand this approach being suitable for multi-view video alignment, since there we are fine-tuning it for each task and 3DTRL can build implicit representation, I cannot see this as a generic module for learning viewpoint-agnostic representation.

4. Also, from the depth-map visualization (Fig. 6), I think it is just learning to pay attention to the primary parts of the class (e.g. for each of the dog images, the eyes, ears, and noses have very low value). I cannot see the depth being learned here.

5. Authors mention that this is a generic plug-and-play (L14, L48), but all the experiments are only with DeiT. More experiments with different architecture are needed to support the claim.

The notations and math are also unclear at many places:
1. E.g. in eq (2) authors represent (u,v) as center pixel coordinates. While in eq (3), (u,v) are token 2D locations.

2. The camera model presented in eq (2) is not a standard pinhole camera model. In standard pinhole camera if dn is a depth, then xn = dn*un/c , yn = dn*vn/c , zn=d (considering camera frame and image frame matches at (0,0))

Some typos/writing mistakes

L55: (u, v) are pixel coordinates: (u, v) are not any pixel coordinates, they are coordinates of the center pixel of the image.
L104: which is beneficial for later representation learning → which is beneficial later for representation learning (or did you mean, which is beneficial for better representation learning)
L133: raw → yaw
Fig4: 3DTPL → 3DTRL


References:
[1] Li, Xiaotian, Juha Ylioinas, and Juho Kannala. "Full-Frame Scene Coordinate Regression for Image-Based Localization." Robotics: Science and Systems Conference. University of Queensland, 2018.

[2] Brachmann, Eric, and Carsten Rother. "Learning less is more-6d camera localization via 3d surface regression." Proceedings of the IEEE conference on computer vision and pattern recognition. 2018.

[3] Liu, Ze, et al. "Swin transformer: Hierarchical vision transformer using shifted windows." Proceedings of the IEEE/CVF International Conference on Computer Vision. 2021.

---

> ### Author Response · Authors · 2022-08-02
> **Authors' Response to Reviewer 6XtT - Part 1**
>
> We appreciate the reviewer for the thoughtful comments and contrusctive suggestions.
>
> - > Camera estimation
>   * We believe our estimated extrinsics are doing object-centric canonicalization of images with respect to their object poses. This aligns with the Reviewer **xnSs**’s insight: "I think we expect the poses (extrinsics) are doing some canonicalization of the input imagery -- registering them closer to a common pose."
>   Our new visualization (**Figure 10** in the updated supplementary material) also implicitly illustrates this by showing that the images with the same/similar object pose (in different environments) result in similar camera extrinsics regardless of the background.
>
> - > Limitation on ImageNet-perturbed Evaluation
>   * We agree that perspective transformation on ImageNet is not exactly true viewpoint change, and it is more a proof-of-concept. Simultaneously, the paper also has experimental results on real-world (1) multi-view video alignment and (2) cross-view action recognition tasks. Our approach shows meaningful improvements in these tasks which was also mentioned by Reviewers **xnSs, FjVH, sPDR, and kBS4**.
>   * Following the suggestion from the reviewers, we further tested the model performance on ObjectNet [A1], which is a very challenging real-world testset including viewpoint and other distracting changes. Our new experiments show that our method consistently and meaningfully outperforms its corresponding baseline model. Note that we are using the ImageNet-trained model without any fine-tuning for ObjectNet.
> 	| Model        | ObjectNet         |
> 	|--------------|-------------------|
> 	| DeiT-T       | 21.30             |
> 	| DeiT-T+3DTRL | **22.37 (+1.07)** |
> 	||
> 	| DeiT-S       | 25.83             |
> 	| DeiT-S+3DTRL | **27.08 (+1.25)** |
> 	||
> 	| DeiT-B       | 26.98             |
> 	| DeiT-B+3DTRL | **27.34 (+0.36)** |
> 	||
> 	| Swin-T       | 28.60             |
> 	| Swin-T+3DTRL | **28.95 (+0.35)** |
> 	||
> 	| Swin-S       | 30.85             |
> 	| Swin-S+3DTRL | **31.26 (+0.41)** |
>
> - > Generic module
>   * We would like to clarify. What we mean is that 3DTRL could be inserted within the model for different downstream tasks that would benefit from 3D, as long as we have training data for these downstream tasks. 3DTRL will be optimized for the given dataset. Our new experiments also show the potential that the learned 3DTRL could transfer from ImageNet to ObjectNet without any finetuning, and we will investigate this further in the final version of the paper.
>
> - > Focus on primary parts of the object in pseudo-depth estimation
>   * In order to clarify this further, we included additional depth map visualizations in the updated supplementary material. **Figure 13** shows more examples with multiple objects in the scene (i.e., we have other objects in the scene). In these examples, we observe that foreground objects that do not correspond to the object class label also provide reasonable pseudo-depth values.
>   * We also conducted an additional experiment quantitatively measuring the accuracy of the 3D depth estimation with the dataset where the ground truth 3D surfaces are known (**Figure 8** in the updated supplementary material). The result shows that our pseudo-depth estimation is highly correlated with the ground truth depth maps.
>
> [A1] Barbu, Andrei, et al. "Objectnet: A large-scale bias-controlled dataset for pushing the limits of object recognition models." Advances in neural information processing systems 32 (2019).

---

> > ### Comment · Reviewer_6XtT · 2022-08-07
> > **Further organization is needed**
> >
> > Authors have provided detailed response to all of my concerns. I saw that many reviewers raised similar concerns and these are important pieces of information for the paper to be accepted. As reviewer xNss mentioned, the common concerns across all reviewers should me moved to the main paper, I understand this is a bit of work towards the rebuttal, but this can make paper really strong. Especially, depth correlation analysis and showing improvement on different transformer backbones.
> >
> > Thank You.

---

> > > ### Author Response · Authors · 2022-08-08
> > > **Material Organization**
> > >
> > > We thank the reviewer for recognizing the significance of new experiments. We totally agree that important evaluations should appear in the main paper, and we will do so in the camera-ready version by utilizing an additional content page. We have attempted a replica of this 10-page camera-ready version at the end of the Appendix (beginning from Page 13 in the newly updated supplementary) showing our reorganization.

---

> > > > ### Comment · Reviewer_6XtT · 2022-08-09
> > > > **Thank You**
> > > >
> > > > Dear Authors,
> > > >
> > > > Thanks for a reminder. I think 10 page version looks good. One suggestion is for Fig.8, please add object class on top.
> > > >
> > > > Question about Fig. 6 and Fig. 7, I see in figure 6 x-axis goes till epoch 300, but in Fig. 7 only till epoch 70. Is there a reason for this? Ideally both figures should have same x-axis.
> > > >
> > > > And since, I see 3D information is being learned pretty early on in the training process, if possible it would be good to add depth-map visualization at different training stages for a few images in the appendix. I am wondering if depth is changing at all afterwards.

---

> > > > > ### Author Response · Authors · 2022-08-09
> > > > > **Thanks and please check the updated supplementary**
> > > > >
> > > > > Thanks for the suggestions. We have some updates and please check the 10-page version in the updated supplementary to see our changes.
> > > > >
> > > > > > Add object class on top for Figure 8
> > > > >   - We have added class labels in the updated Figure 8.
> > > > >
> > > > > > Epochs
> > > > >   - We apologize for the confusion, they are from different training datasets (Figure 6: IN-1K; Figure 7: Can dataset).
> > > > >
> > > > > > Pseudo-depth map at different stages
> > > > >   - We have attached pseudo-depth estimation results at epoch 10, 40, 200 and 300 of training on IN-1K, in **Section K, Figure 15** (the last page of the updated supplementary). We find that the estimation varies significantly from epoch 10 to epoch 40 (higher foreground-background correctness, less missing parts of objects), but changes only a bit from epoch 40 to epoch 200 and finally to epoch 300 (mostly scales). This observation is also coherent with our quantitative evaluation. Thus, the pseudo-depth estimation learns promptly, however the model convergence takes longer time since we are optimizing for a downstream task (eg. classification).

---

> > > ### Author Response · Authors · 2022-08-09
> > > **Reminder**
> > >
> > > It will be great if the reviewer can please check the 10-page final version of the main paper we attached at the end of the newly updated supplementary (from **Page 13**). This version reflects the suggested paper organization by the reviewers and we would be happy to get further comments.
> > >
> > > Please let us know if we have successfully addressed your concerns. If so, we will be very pleased if it can be reflected in your final review.

---

> ### Author Response · Authors · 2022-08-02
> **Authors' Response to Reviewer 6XtT - Part 2**
>
> - > Experiment with more Transformer architectures
>   * In **Section 4.5** of the submitted paper, we showed the results with TimeSformer, which invokes a different input (video) and different attention mechanism (divided space-and-time) than DeiT.
>   * In order to further confirm this, following the suggestion from the reviewer, we newly conducted experiments by inserting 3DTRL with different architectures like Swin [A2] and TnT [A3]. The table below shows the results. We find that the improvement in Swin is relatively smaller compared to the other Transformers, due to the strong inductive bias (local windows) that limits the interaction among tokens. Still, the result is consistent across different architectures.
> 	| Model          | CIFAR-10         | CIFAR-100        | Pouring (Kendall’s Tau) | Pick(Kendall’s Tau) |
> 	|----------------|------------------|------------------|-------------------------|---------------------|
> 	| Swin-T         | 50.11            | 21.53            | 0.584                   | 0.623               |
> 	| Swin-T + 3DTRL | **50.29(+0.18)** | **21.55(+0.02)** | **0.683(+0.099)**       | **0.640(+0.017)**   |
> 	||
> 	| TnT-S          | 81.25            | 54.07            | 0.740                   | 0.640               |
> 	| TnT-S + 3DTRL  | **82.43(+1.18)** | **56.00(+1.93)** | **0.792(+0.052)**       | **0.671(+0.031)**   |
>
> - > Experiment with perspective augmentation
>   * Following the suggestion, we applied perspective augmentation during the training, focusing on the multi-view video alignment task. We find that it does not help the multi-view video alignment. Instead, it harms the training since these augmentation transforms are not true viewpoint changes. For simplicity we show one metric Kendall’s tau (the higher the better) in the table below.
>
> 	|                      | Pouring | Pick   | MC     | Can   | Lift  |
> 	|----------------------|---------|--------|--------|-------|-------|
> 	| DeiT                 | 0.426   | 0.245  | -0.115 | 0.789 | 0.716 |
> 	| DeiT+Perspective Aug | 0.201   | -0.249 | -0.419 | 0.342 | 0.486 |
> 	| DeiT+3DTRL           | 0.740   | 0.635  | 0.392  | 0.824 | 0.739 |
>
>
> - > Eq(2)
>   * Good catch. Sorry for the confusion. (u, v) in eq (2) are center pixel coordinates, which are set to (0,0). We have fixed it in the revised version of the paper.
>
> - > Eq(3)
>   * Sorry for the confusion. Both versions are mathematically correct and the difference is the definition of $d$. If $d$ is the distance between the point and the optical center, then $z=dc/\sqrt{u^2 + v^2 + c^2}$. We missed $c$ in the nominator and $c^2$ (originally written in $c$) in the denominator, but our results are not affected since $c$ is set to 1 in our configuration. We have fixed this equation using the standard way.
>
> [A2] Liu, Ze, et al. "Swin transformer: Hierarchical vision transformer using shifted windows." Proceedings of the IEEE/CVF International Conference on Computer Vision. 2021.
>
> [A3] Han, Kai, et al. "Transformer in transformer." Advances in Neural Information Processing Systems 34 (2021): 15908-15919.

---

### Official Review · Reviewer_sPDR · 2022-07-11

**Rating:** 7
**Confidence:** 4
**Soundness:** 3 good
**Presentation:** 3 good
**Contribution:** 3 good

**Summary:**

This paper proposes 3DTRL, a 3D-aware component that can be plugged into transformer-based supervised/selfsupervised learners. In particular, 3DTRL utilizes the tokens generated by the previous layers, and uses per-token MLP to predict psuedo depth values for each token. Then, a MLP-based camera pose estimator uses all the tokens to predict the euler angles and camera translation. In the end, a 3D embedding is used to generate embeddings for each token given their predicted camera-space coordinates. The following transformer layers will therefore be 3D aware.
To support the validity of this proposed module, extensive experiments are conducted. The three main tasks are image classification under prespective transform, video alignment under different viewpoints, and activity classification. The proposed module only induces slight increase in parameter counts and memory usage, but improves the final result by a sigificant margin.

**Questions:**

1. I simply find it interesting that imposing a structural prior is already effective with a significant margin. Does the choice of camera pose parameterization affect the results at all? Euler-angles have gimbal locks and therefore induce ambiguity, which might be harmful.

2. The camera pose results visualized in the supp. is quite interesting. Even though it might be a strech, is it possible to compare how well the camera poses improved over the course of training? As argued in the paper, precise depth and camera pose is not required to make good classification decision, but how well shoud they be to make the 3D embedding meaningful?

**Limitations:**

The authors do not discuss the limitation of the method. It is important to have a up-front discussion of the limitations, and potential bells and whistles.

**Strengths And Weaknesses:**

Strengths
1. Originality:

The proposed module is novel from the reviewer's perspective. It's simple in nature, imposing camera transformations as a structual prior into the transformers.

2.Quality:

I find this paper of good quality. It presents all the necessary details and experiements to support the calim, and addresses the design choices well through ablation studies.

3. Significance:
I find this paper significant, and hopefully to the community as well. The idea itself is simple, yet proves to be effective. In addition to the applications showed in this paper, it may also be helpful for areas such as unsupervised depth/pose estimations, especially how they could benefit from large image/video datasets without annotations.

4. Clarity:
I find this paper easy to follow and well written.


Weakness:
This is not a major issue, but I'm not convinced that the predicted depth/camera poses are meaningful in terms of explaing the image geometrically. The circular tokens in Fig.1 doesn't seem to expain the image, and there's less visualization on the predicted pose. I do not find this very concerning though, since the prior is really weak as they are only structrual in nature.

---

> ### Author Response · Authors · 2022-08-02
> **Authors' Response to Reviewer sPDR**
>
> We thank the reviewer for the valuable comments.
> - > Depth/Camera pose evaluation
>   * Following the suggestion from the reviewers, we conducted additional experiments to quantitatively evaluate the predicted 3D locations of the features as well as the camera poses. **Section A** in Appendix (in the updated supplementary material) shows the results. The accuracy of the predicted depth map in each image was evaluated by measuring the correlation to the ground truth. The resulting correlation coefficient $r$ is ~0.7, showing that we have a good mapping.
>   * In evaluation on camera poses, we test the disparity between the estimation and the ground truth. We individually check the disparity of position and orientation, and results (both <0.5) show that our estimation is fairly correlated to the ground truth.
> - > Gimbal lock
>   * Thanks for the great question. We agree that gimbal lock is an important issue in 3D rotations. However, in our case, the Euler-angles are predicted by neural nets so that the gimbal lock case occurs rarely compared to the real continuous controlling process.
> - > Optimization over the course of training
>   * We conducted additional experiments to further investigate these. Please check **Section A** (**Figure 8** and **9**) in the updated supplementary for the performance curve.

---

> > ### Comment · Reviewer_sPDR · 2022-08-07
> > **Thanks for the udpates**
> >
> > Thank you for the updates and answers!
> >
> > This is promising, and I would like to raise the concern that asking NNs to generate euler angles will induce wierd behavior[1][2], though not really a concern of this paper.
> >
> > [1] On the Continuity of Rotation Representations in Neural Networks, CVPR 2019
> > [2] Eliminating topological errors in neural network rotation estimation using self-selecting ensembles, TOG 2021

---

> > > ### Author Response · Authors · 2022-08-08
> > > **Rotation Representations**
> > >
> > > We thank the reviewer for sharing two solid papers regarding rotation representations in neural networks. As mentioned, it was not a big problem in our case as the approach probably learned to avoid putting R into nearly boundary cases. We will add the discussions in the final version of the paper.

---

### Official Review · Reviewer_FjVH · 2022-07-12

**Rating:** 7
**Confidence:** 3
**Soundness:** 3 good
**Presentation:** 4 excellent
**Contribution:** 2 fair

**Summary:**

The paper proposes 3DTRL, a module in ViT which helps the learning of viewpoint-agnostic representations. After ViT extracts image patches to tokens, 3DTRL estimates a depth for the whole patch, as well as camera extrinsics. Then the features are transformed into 3d tokens, and feed into the next transformer layers. Experiments are conducted on both image classification and multi-view video alignment. 3DTRL improves the performance on more viewpoints.

**Questions:**

- Is it possible to evaluate the learned camera extrinsics R and T? At least in simulators the ground truth camera pose should be available.
- Do you notice any failure modes introduced by 3DTRL?
- The paper writes the focal length c is set as a constant hyperparameter. What do you finally use? I guess you might use different c for different datasets since c is available for synthetic datasets?

**Limitations:**

Limitations are not discussed in the paper.

**Strengths And Weaknesses:**

Strengths:

- It is a neat idea to add an additional module to transformer so that transformer can learn viewpoint-agnostic features. Pseudo depth estimation on patches makes a lot of sense.
- My primary concern after reading the idea is whether pseudo depth estimation and camera pose estimation can really learn any meaningful information. Especially, the pseudo depth estimation is just 2 MLP layers. And it is addressed well in Fig 6 and supp Fig 7. I appreciate it!


Weaknesses:

- The pseudo-depth prediction seems to have scale ambiguity -- how does the model learn metric depth without supervision? Do you enforce scale-invarance somewhere?
- The learned camera pose also seems to have ambiguity. Take Fig 7 as an example. It seems multiple camera trajectories are reasonable in both cases. We only care about the relative camera pose so the absolute camera pose does not matter. I'm not sure how networks learn an absolute camera pose in this case.
- The improvement on large datasets such as ImageNet is relatively small (around 0.2%). It probably because ImageNet is too diverse and does not benefit from viewpoint-agnostic representations. At the same time, it works well on ImageNet-perturbed. However, these perturbed images may have additional cues to learn geometric transformations based on the black boundary.

---

> ### Author Response · Authors · 2022-08-02
> **Authors' Response to FjVH**
>
> We thank the reviewer for the valuable comments.
> - > Scale ambiguity
>   * We agree with the reviewer that there is scale ambiguity. We do not enforce any scale-invariance, but optimize the entire model including 3DTRL with respect to the downstream task (e.g., object classification). The intuition is that 3DTRL has to learn to set the pseudo-depth at the right scale, in order to make the entire task (e.g., object recognition) successful. The resulting 3D estimation will not be metric (as it is not explicitly optimized to recover the ground truth 3D) but have its own scale, and we find that this is ok as long as it is consistent across different images.
>   * We conducted a new experiment evaluating the correlation between the estimated pseudo-depth and the ground truth depth (**Section A.1** in the updated supplementary material), and confirmed that they are correlated (r is ~0.7).
>
> - > Ambiguity in camera poses
>   * As we described above, we believe the 3DTRL is optimized to learn camera parameters setting the pseudo-depth at the right scale to make the task successful. In the new experiment evaluating the camera estimation (**Section A.2**), we show a fair alignment (disparity < 0.5) between our camera estimation and the ground truth.
>
> - > Limitation on ImageNet-perturbed evaluation
>   * We agree that the perspective transformation may not represent reality. The perturbed ImageNet is a good proof-of-concept, and we also evaluated 3DTRL on true multi-view scenarios such as video alignment and action recognition (Section 4.2, 4.4 in the main paper).
>   * We also conducted one extra experiment with ObjectNet [A1], a very challenging test set including viewpoint and other distracting changes, compared to ImageNet. We show that our method consistently outperforms its corresponding baseline model, including the additional Swin Transformer backbone.
> 	| Model        | ObjectNet         |
> 	|--------------|-------------------|
> 	| DeiT-T       | 21.30             |
> 	| DeiT-T+3DTRL | **22.37 (+1.07)** |
> 	||
> 	| DeiT-S       | 25.83             |
> 	| DeiT-S+3DTRL | **27.08 (+1.25)** |
> 	||
> 	| DeiT-B       | 26.98             |
> 	| DeiT-B+3DTRL | **27.34 (+0.36)** |
> 	||
> 	| Swin-T       | 28.60             |
> 	| Swin-T+3DTRL | **28.95 (+0.35)** |
> 	||
> 	| Swin-S       | 30.85             |
> 	| Swin-S+3DTRL | **31.26 (+0.41)** |
>
> - > Evaluation on the estimated cameras
>   * We conducted additional experiments. Please refer to **Section A.2** in our updated supplementary, where we quantitatively and qualitatively evaluate the estimated extrinsics. We show fair disparities (<0.5) in both position and orientation measurements, indicating that our estimation has a fair correspondence to the ground truth.
>
> - > Failure cases
>   * We find it hard to estimate small objects in the scene, or complex scene, due to the coarse scale (in 16x16 image patches) from the backbone Transformer. More discussion is in the Limitation section in the updated supplementary.
>
> - > Focal length c
>   * We set $c=1$ for simplicity. Given by eq.(3), $d$ and $c$ are correlated so estimating $d$ and when $c$ is fixed is enough. For different datasets, we don’t change $c$, i.e. they are all set to $1$.
>
> - > Ground truth camera parameters for training
>   * Our method does not use any ground truth intrinsics and extrinsics for training.
>
>
> [A1] Barbu, Andrei, et al. "Objectnet: A large-scale bias-controlled dataset for pushing the limits of object recognition models." Advances in neural information processing systems 32 (2019).

---

> > ### Comment · Reviewer_FjVH · 2022-08-08
> > **Re: Authors' Response to FjVH**
> >
> > Thank you for your updates! I'm quite impressed by the updated experiments (especially the evaluation of depth and camera pose). That's a lot of efforts. In general, I'm convinced by additional experiments. I'm happy to raise my rating from 6 to 7.
> >
> > Just some suggestions: The camera pose evaluation can be done using accuracy/AUROC/median angle error. That might be more straightforward to interrupt. [1] and [2] might be inspiring in light of their evaluation of camera poses.
> >
> > [1] Sarlin et al. SuperGlue: Learning Feature Matching with Graph Neural Networks. CVPR 2020.
> > [2] Jin et al. Planar Surface Reconstruction from Sparse Views. ICCV 2021.
> >
> > Depth also has scale-invariant RMSE but I'm not sure if you can get any meaningful results here since it focus a lot on details. We don't expect the implicit learning of depth gets most details correctly here.
> >
> > Anyway, thanks for your great work!

---

### Official Review · Reviewer_xnSs · 2022-07-12

**Rating:** 6
**Confidence:** 4
**Soundness:** 3 good
**Presentation:** 3 good
**Contribution:** 3 good

**Summary:**

This paper presents a method for transformers to upgrade their 2D image inputs to pseudo-3D, and shows quite convincingly that this contributes a performance boost. The basic idea is a layer that goes somewhere in the middle of the transformer: estimate per-token depth and also a camera pose (pitch yaw roll), then unproject coordinates to 3D, encode these coordinates into embeddings, then add these with the existing embeddings, and proceed with the rest of the transformer. This gives performance boosts on a variety of tasks.


**Questions:**

When you use multiple 3DTRL locations (in the supp), do you share the depth and pose estimates across all of them?


**Limitations:**

I think the authors forgot to write a limitations section, but I think they could resolve this easily.

**Strengths And Weaknesses:**

I am quite amazed by this paper. The idea is simple and intuitive, it adds negligible computation, it requires no extra supervision, and yet it improves results by a few points on a variety of tasks. The ablation studies indicate that it is indeed the 3d lifting, and not the additional parameters, that leads to the performance benefit. I think this is a significant result and worth publishing. Also the paper is easy to read.

As for weaknesses: I wish the paper had a more thorough analysis, even if qualitative, to show why exactly this is working. The depth maps of the dogs in Figure 6 are somewhat helpful, but why so few, and why just dogs? I would like to see a random sample of depth maps, to get a feel for the normal behavior. (I found a few more in the supplementary, but for me the 3D visualizations there are less clear than the depth maps.) Also, analysis of the camera poses seems to be completely missing. I may have missed it, but I'm not sure the pitch/yaw/roll estimation was shown to be helping. Maybe the model is only estimating some basic "intrinsics", to help with the scaling as the points go to 3D. I think we expect the poses (extrinsics) are doing some canonicalization of the input imagery -- registering them closer to a common pose. Besides adding the basic ablations, it would be great, for example, to get a dataset with pose annotations, do some self-supervised learning, and then show that the estimated camera poses have some mapping (even if weak) to the real pose distribution.

---

> ### Author Response · Authors · 2022-08-02
> **Authors' Response to Reviewer xnSs**
>
> We thank the reviewer for the encouragement and the insightful comments.
> - > More (depth map) visualizations
>   * Following the suggestion, we added more depth map examples in **Figure 14** in the updated supplementary material. We show examples of multiple object classes.
>
> - > "I think we expect the poses (extrinsics) are doing some canonicalization of the input imagery -- registering them closer to a common pose."
>   * We agree with the reviewer. In order to confirm this further, we conducted a qualitative analysis by taking images of the same object with different viewpoints (and different background), using our ImageNet trained model. **Figure 10** in the updated supplementary material illustrates the results. We are able to observe that the images containing similar object poses result in similar camera poses, implicitly suggesting that the estimated extrinsics canonicalise the object pose.
>
> - > 3D estimation and camera pose analysis
>   * We also added quantitative evaluation regarding 3D estimation (pseudo-depth) and camera pose estimation in **Section A** in the updated supplementary.
>   * As our estimations and the ground truth do not share the same coordinate system (and as our model only optimizes with respect to the entire task, eg. classification), we measure the relative mapping between estimation and ground truth.
>   * We showed that the correlation between our pseudo-depth and ground truth is high (~0.7). The disparity between estimated camera poses and ground truth camera poses is fair (disparity < 0.5). These suggest 3DTRL is able to relatively recover 3D information.
>
> - > Question: "When you use multiple 3DTRL locations (in the supp), do you share the depth and pose estimates across all of them?"
>   * The depth and pose estimates are not shared.

---

> > ### Comment · Reviewer_xnSs · 2022-08-05
> > **Great**
> >
> > Great, these additions answer my questions and improve the paper. It seems other reviewers shared a similar concern, about getting a better intuition of why this works, and I think the additional visualizations and analysis are a great help on this matter.
> >
> > This will probably take a big effort, but: I think the main paper could be revised to do a better job of addressing concerns as they pop up in the reader's mind. Shifting a few good plots and visualizations from the supplementary into the main, while making the text more concise, would make the paper much easier to like.

---

> > > ### Author Response · Authors · 2022-08-05
> > > **Improving Paper Organization**
> > >
> > > We thank the reviewer for the suggestion on improving the paper organization. We will move suggested content from the supplementary to the main paper accordingly in the final version.

---

> > > > ### Author Response · Authors · 2022-08-08
> > > > **10-page Final Version Attached**
> > > >
> > > > As the reviewers suggested, we have attempted a replica of 10-page final version at the end of the Appendix (beginning from Page 13 in the newly updated supplementary) showing our reorganization. We would appreaciate any comments from you about the reorganization.

---

### Author Response · Authors · 2022-08-02
**Summary of New Contents**

We thank all the reviewers for the thoughtful comments. As reviewers suggested, we conducted new experiments and included more examples in the **updated supplementary material**. Here is the summary.
1. [Section **A.1**] Quantitative evaluation on pseudo-depth estimation with ground truth depth maps.
2. [Section **A.2**] Quantitative and qualitative evaluations on camera estimation with ground truth camera extrinsics.
3. [Section **B**] Evaluation on image classification using ObjectNet, a dataset including hard, real-world image samples in different rotations and viewpoints.
4. [Section **C**] Evaluation on 3DTRL with more Transformer architectures.
5. [Section **D**] Comparison between naive perspective augmentation and 3DTRL.
6. [Section **F**] Examples for pseudo-depth estimation on non-class objects.
7. [Section **H**] Limitation discussion.
8. [Section **M**] A large collection of pseudo-depth map examples.

---

### Meta-Review · Area_Chair_eyfN · 2022-08-25

**Recommendation:** Accept
**Confidence:** Less certain

**Metareview:**

This paper presents a method for transformers to upgrade the 2D image input to pseudo-3D. It proposes a neural layer that estimates per-token depth and also a camera pose (pitch yaw roll), then unproject token coordinates to 3D, encodes these coordinates into embeddings, then adds these with the existing embeddings, and proceeds with the rest of the transformer. This gives performance boosts on a variety of tasks, such as video alignment.
The reviewers raised concerns regarding the depth maps inferred looking more like saliency maps, but also, the depth scale ambiguity, and the ambiguity of absolute camera pose inference. The rebuttal submitted by the authors included additional results that showed that the inferred depth maps and camera poses correlated with the correct ones. All reviewers appreciated the additional experiments contributed by the authors, and suggested them to be included to the main paper.


**Award:**

No

---

### Decision · Program_Chairs · 2022-09-14

Accept